# BranchGRPO: Stable and Efficient GRPO with Structured Branching in Diffusion Models

**Yuming Li**[1]* **Yikai Wang**[2]* **Yuying Zhu**[3]
**Zhongyu Zhao**[1] **Ming Lu**[1] **Qi She**[3] **Shanghang Zhang**[1]†
[1]State Key Laboratory of Multimedia Information Processing,
School of Computer Science, Peking University
[2]Beijing Normal University  [3]ByteDance
shanghang@pku.edu.cn

## Abstract

Recent progress in aligning image and video generative models with Group Relative Policy Optimization (GRPO) has improved human preference alignment, but existing variants remain inefficient due to sequential rollouts and large numbers of sampling steps, unreliable credit assignment, as sparse terminal rewards are uniformly propagated across timesteps, failing to capture the varying criticality of decisions during denoising. In this paper, we present BranchGRPO, a method that restructures the rollout process into a branching tree, where shared prefixes amortize computation and pruning removes low-value paths and redundant depths. BranchGRPO introduces three contributions: (1) a branching scheme that amortizes rollout cost through shared prefixes while preserving exploration diversity; (2) a reward fusion and depth-wise advantage estimator that transforms sparse terminal rewards into dense step-level signals; and (3) pruning strategies that cut gradient computation but leave forward rollouts and exploration unaffected. On HPSv2.1 image alignment, BranchGRPO improves alignment scores by up to **16%** over DanceGRPO, while reducing per-iteration training time by nearly **55%**. A hybrid variant, BranchGRPO-Mix, further accelerates training to 4.7× faster than DanceGRPO without degrading alignment. On WanX video generation, it further achieves higher motion quality reward with sharper and temporally consistent frames.

## 1 Introduction

Diffusion and flow-matching models have advanced image and video generation with high fidelity, diversity, and controllability (Ho et al., 2020; Lipman et al., 2022; Liu et al., 2022). However, large-scale pretraining alone cannot ensure alignment with human intent, as outputs often miss aesthetic, semantic, or temporal expectations. Reinforcement learning from human feedback (RLHF) addresses this gap by directly adapting models toward human-preferred outcomes (Ouyang et al., 2022).

Within RLHF, Group Relative Policy Optimization (GRPO) has shown strong stability and scalability across text-to-image and text-to-video tasks (Liu et al., 2025a; Xue et al., 2025). However, when applied to diffusion and flow-matching models, current GRPO variants still face two fundamental bottlenecks: (1) **Inefficiency.** Standard GRPO adopts a sequential rollout design, where each trajectory must be independently sampled under both the old and new policies. This incurs $O(N \cdot T)$ complexity with denoising steps $T$ and group size $N$, leading to significant computational redundancy and limiting scalability in large-scale image and video generation tasks. (2) **Sparse rewards.** Existing methods assign a single terminal reward uniformly across all denoising steps, neglecting informative signals from intermediate states. This uniform propagation leads to unreliable credit assignment and high-variance gradients, raising the central question: how can we attribute sparse outcome rewards to the specific denoising steps that truly shape final quality?

---

*Equal contribution.
†Corresponding author.

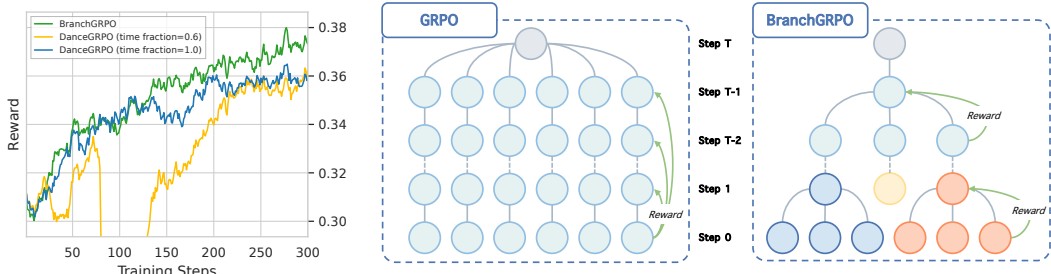

Figure 1: **Comparison of BranchGRPO and DanceGRPO.** *Left:* Reward curves during training. BranchGRPO converges substantially faster, achieving up to **2.2× speedup** over DanceGRPO (time fraction = 1.0) and **1.5× speedup** over DanceGRPO (time fraction = 0.6), while ultimately surpassing both baselines. The time fraction = 0.6 variant further exhibits pronounced instability. (time fraction denotes the proportion of diffusion timesteps used during training.). *Right:* Illustration of rollout structures. GRPO relies on sequential rollouts with only final rewards, whereas Branch-GRPO performs branching at intermediate steps and propagates dense rewards backward, enabling more efficient and stable optimization.

To overcome these limitations, we propose **BranchGRPO**, a tree-structured policy optimization framework for diffusion and flow models. BranchGRPO replaces inefficient independent sequential rollouts with a branching structure, where scheduled *split steps* in the denoising process allow each trajectory to stochastically expand into multiple sub-trajectories while reusing shared prefixes. This design amortizes computation across common segments and aligns naturally with the stepwise nature of denoising, substantially improving sampling efficiency while reducing computational cost.

The tree structure further enables a novel reward fusion with depth-wise advantage estimation. Instead of uniformly propagating a single terminal reward, BranchGRPO aggregates leaf rewards and propagates them backward with depth-wise normalization, producing finer-grained step-level advantages. In addition, width- and depth-pruning strategies remove redundant branches and depths during backpropagation, accelerating training and reallocating computation toward promising regions of the trajectory space.

We validate BranchGRPO on both text-to-image and image-to-video alignment tasks, demonstrating its effectiveness and generality across modalities. In addition, we verify the scaling law of Branch-GRPO, larger group sizes consistently lead to better alignment performance.

Our contributions are threefold:

- We introduce **BranchGRPO**, a *tree-structured GRPO training paradigm*. It replaces independent sequential rollouts with branching during denoising, reusing shared prefixes to amortize compute and broaden exploration, thereby improving efficiency and scalability.

- We propose a new reward fusion and depth-wise advantage estimation method that converts sparse terminal rewards into dense, step-level signals, yielding more stable optimization.

- We design complementary width- and depth-pruning strategies that lower backpropagation cost and further improve alignment.

## 2 RELATED WORK

Diffusion models (Ho et al., 2020; Rombach et al., 2022) and flow matching models (Lipman et al., 2022; Liu et al., 2022) have become dominant paradigms for visual generation due to their strong theoretical foundations and ability to generate high-quality content efficiently. While pretraining establishes the generative prior, aligning outputs with nuanced human preferences requires reinforcement learning from human feedback (RLHF). In natural language processing, RLHF has proven highly successful for aligning large language models (LLMs) (Ouyang et al., 2022; Christiano et al., 2017; Lu et al., 2025), where methods such as PPO and GRPO enable stable preference-driven post-training. These successes have inspired adaptation of RLHF to vision.

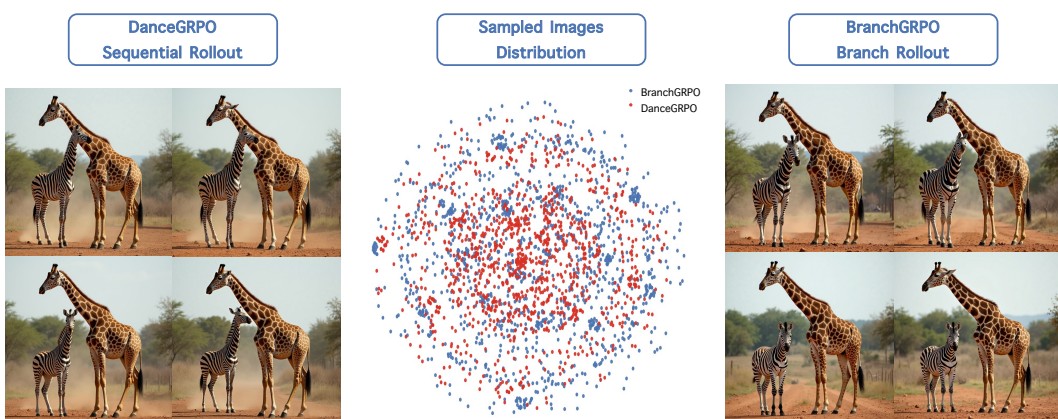

Figure 2: Comparison of sequential and branch rollouts. **Left/Right:** example generations from DanceGRPO and BranchGRPO, respectively. **Middle:** distribution of sampled images projected into 2D feature space, where red and blue dots correspond to DanceGRPO and BranchGRPO.

In the visual domain, RLHF for diffusion has been developed along two main directions (Oshima et al., 2025). Reward-model-based approaches such as ImageReward (Xu et al., 2023) back-propagate learned rewards through the denoising process. Direct Preference Optimization (DPO) (Rafailov et al., 2023; Liu et al., 2025c) has also been extended to diffusion, leading to Diffusion-DPO (Wallace et al., 2024) and Videodpo (Liu et al., 2025c), which achieve competitive alignment without explicit reward modeling. Policy-gradient formulations such as DDPO (Black et al., 2023) and DPOK (Fan et al., 2023) further explore online optimization but often face stability challenges. Meanwhile, standardized reward models including HPS-v2.1 (Wu et al., 2023), VideoAlign (Liu et al., 2025b) enable systematic comparison of alignment algorithms on image and video tasks.

More recently, Group Relative Policy Optimization (GRPO) (Shao et al., 2024) has been introduced as a scalable alternative to PPO for preference optimization. DanceGRPO (Xue et al., 2025) and Flow-GRPO (Liu et al., 2025a) pioneers the application of GRPO to visual generation, unifying diffusion and flow models via SDE reformulation and demonstrating stable optimization across text-to-image, text-to-video, and image-to-video tasks. TempFlow-GRPO (He et al., 2025) further highlights the limitation of sparse terminal rewards with uniform credit assignment, proposing temporally-aware weighting across denoising steps. MixGRPO (Li et al., 2025a) further enhances efficiency with a mixed ODE–SDE sliding-window scheme, though it still faces trade-offs between overhead and performance. Our work continues this line by introducing BranchGRPO, which leverages branching rollouts, depth-wise reward fusion, and structured pruning to improve both stability and efficiency; while related to TreePO in LLMs (Li et al., 2025b), our method adapts tree-structured rollouts specifically to diffusion dynamics.

## 3 METHOD

### 3.1 DOES BRANCH ROLLOUT HARM DIVERSITY?

A natural concern with branch rollouts is that reusing shared prefixes might reduce sample diversity. To investigate this, we generate 4096 samples each from DanceGRPO and BranchGRPO and evaluate their distributions across multiple feature spaces. Figure 2 provides both qualitative and quantitative evidence: the left/right panels show representative generations, while the middle panel visualizes the sampled distributions in a 2D feature embedding, where DanceGRPO and Branch-GRPO points largely overlap.

Quantitatively, in the Inception feature space, the distributions remain close, with KID (Bińkowski et al., 2018)=0.0057 and $\text{MMD}^2$ (Gretton et al., 2012)=0.0067. In the CLIP feature space (ViT-B/32 (Radford et al., 2021)), the similarity is even stronger: KID=0.00022 and $\text{MMD}^2 = 0.0149$, both indicating that the two distributions are almost indistinguishable at the semantic level.

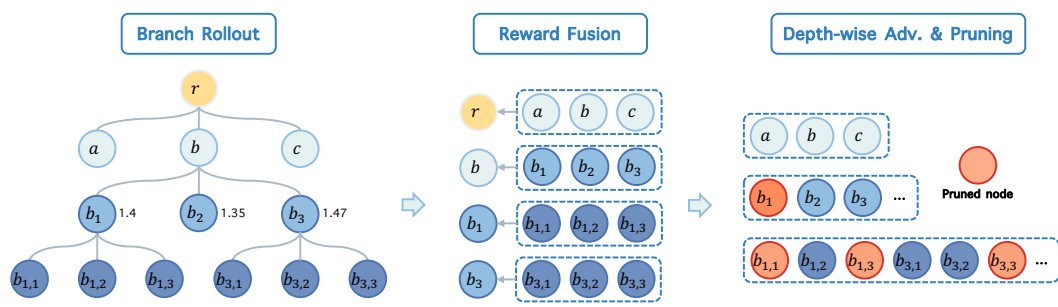

Figure 3: **Left:** branch rollout process . **Middle:** leaf rewards are fused upward. **Right:** depth-wise normalization and pruning yield dense advantages and reduce cost.

Taken together, Figure 2 and these statistics demonstrate that branch rollouts preserve distributional and semantic diversity, introducing at most negligible shifts across different feature spaces.

## 3.2 BRANCH ROLLOUT ALGORITHM

**Preliminaries.** Given a prompt, BranchGRPO reformulates denoising into a tree-structured process. We align terminology with tree search: (i) depth $T$ denotes the number of denoising steps; (ii) width $w$ is the number of completed trajectories (leaves); (iii) branching steps $\mathcal{B}$ indicate split timesteps; (iv) branch correlation $s$ controls the diversity among child nodes; (v) branching factor $K$ is the number of children per split.

**Branch sampling.** Unlike prior GRPO variants such as DanceGRPO and FlowGRPO that rely on *sequential rollouts*, where each trajectory is sampled independently from start to finish, Branch-GRPO reorganizes the process into a tree-structured rollout (Figure 1). For each prompt, we initialize a root node with a same initial noise $z_0 \sim \mathcal{N}(0, I)$ and then denoise step by step along the reverse SDE. At designated split steps $\mathcal{B}$, the current state expands into $K$ children, producing multiple sub-trajectories that share early prefixes but diverge afterward. The branching is achieved by injecting stochastic perturbations into the SDE transition, with a hyperparameter $s$ controlling the diversity strength among child nodes. This mechanism balances exploration diversity and stability while keeping the marginal distribution unchanged. The rollout continues until reaching the maximum depth $T$, at which point all leaves are collected for reward evaluation.

Formally, following Xue et al. (2025), the reverse-time dynamics can be written in the SDE form:

$$\mathrm{d}z_t = \left(f_t z_t - \frac{1+\varepsilon_t^2}{2} g_t^2 \nabla \log p_t(z_t)\right) \mathrm{d}t + \varepsilon_t g_t \, \mathrm{d}w_t, \tag{1}$$

where $\varepsilon_t$ controls stochasticity.

At a split step $i \in \mathcal{B}$ with step size $h_i = t_i - t_{i+1}$, instead of sampling a single successor we generate $K$ correlated children:

$$z_{i+1}^{(b)} = \mu_\theta(z_i, t_i) + g_{t_i}\sqrt{h_i}\,\xi_b, \qquad \xi_b = \frac{\xi_0 + s\,\eta_b}{\sqrt{1+s^2}}, \quad b = 1, \dots, K, \tag{2}$$

where $\xi_0, \eta_b \overset{\text{i.i.d.}}{\sim} \mathcal{N}(0, I)$, with $\xi_0$ shared across branches and $\eta_b$ denoting branch-specific innovations. The parameter $s \geq 0$ tunes inter-branch correlation: small $s$ yields highly correlated but stable branches, while large $s$ makes branches nearly independent. By construction, each $\xi_b \sim \mathcal{N}(0, I)$, so every child $z_{i+1}^{(b)}$ has the same marginal distribution as the baseline SDE step.

## 3.3 REWARD FUSION AND DEPTH-WISE ADVANTAGE ESTIMATION

Branch rollouts form a trajectory tree with shared ancestral prefixes, allowing internal node values to be expressed by descendant rewards and enabling backward propagation of leaf signals. However, existing GRPO variants use a single terminal reward at every step, ignoring intermediate states and yielding high-variance, unreliable credit assignment (Fig. 3). BranchGRPO addresses this by propagating leaf rewards upward and converting them into dense step-level advantages via path-probability fusion and depth-wise normalization.

**Reward fusion.** We design a dynamically adjustable reward fusion scheme that aggregates leaf rewards into internal node values through a soft weighting mechanism. For an internal node $n$ with descendant leaves $\mathcal{L}(n)$,

$$\bar{r}(n) = \sum_{\ell \in \mathcal{L}(n)} w_\ell^{(n)} r_\ell, \qquad w_\ell^{(n)} = \frac{\exp(\beta s_\ell)}{\sum_{j \in \mathcal{L}(n)} \exp(\beta s_j)}, \quad s_\ell = \log p_{\text{beh}}(\ell \mid n). \tag{3}$$

Here $p_{\text{beh}}$ is the behavior policy and $\beta$ controls concentration. $\beta = 0$ reduces to uniform averaging; when $\beta = 1$, the fusion reduces to weighting by the behavior policy distribution $p_{\text{beh}}(\ell \mid n)$. Uniform averaging is robust to log-prob calibration errors and encourages exploration by retaining low-probability leaves, but introduces variance when branches are many. Path-weighting reduces variance and stabilizes updates, though it may over-concentrate on high-likelihood leaves in deep trees. We empirically compare both variants in Sec. 4.4.

**Depth-wise normalization.** Nodes at the same depth share the same noise level and are thus directly comparable, while rewards across depths vary drastically due to changing noise states. To balance gradient contributions, we normalize aggregated rewards within each depth $d$:

$$A_d(n) = \frac{\bar{r}(n) - \mu_d}{\sigma_d + \epsilon}, \qquad \mu_d = \text{mean}_{n \in \mathcal{N}_d} \bar{r}(n), \ \ \sigma_d = \text{std}_{n \in \mathcal{N}_d} \bar{r}(n), \tag{4}$$

where $\mathcal{N}_d$ denotes all nodes at depth $d$. Each edge advantage $A(e)$ inherits from its child node and is optionally clipped to $[-A_{\max}, A_{\max}]$. This per-depth standardization prevents late denoising steps with smaller variance from dominating, yielding process-dense and balanced credit signals. Compared to GRPO's prompt-level normalization, which broadcasts a single terminal reward, our scheme produces stable gradients and finer credit assignment to the denoising steps that matter.

We optimize the standard clipped GRPO loss over tree edges:

$$J(\theta) = \mathbb{E}\left[\frac{1}{|\mathcal{E}|} \sum_{e \in \mathcal{E}} \min\left(\rho_e(\theta) A(e), \ \text{clip}(\rho_e(\theta), 1 - \epsilon, 1 + \epsilon) A(e)\right)\right], \tag{5}$$

where an edge $e$ denotes a transition $(s_t, a_t)$ at depth $t$, $\mathcal{E}$ is the set of such edges in a behavior tree, and $\rho_e(\theta) = \pi_\theta(a_t \mid s_t)/\pi_{\text{old}}(a_t \mid s_t)$.

## 3.4 Pruning Strategies

While branch rollouts improve efficiency and provide dense process rewards, an excessive number of branches may induce exponential growth in trajectory count, leading to prohibitive backpropagation cost. To further accelerate training, we introduce two complementary pruning strategies in the context of BranchGRPO: **width pruning**, which reduces the number of leaves used for gradient updates, and **depth pruning**, which skips unnecessary denoising steps.

Importantly, pruning is applied only after reward fusion and depth-wise normalization, and affects backpropagation but not forward rollouts or reward evaluation. This design ensures that all trajectories contribute to reward signals, while gradients are computed only for the selected subset.

*Width Pruning.* After computing rewards and advantages for all leaves $\mathcal{L}$, we restrict gradient updates to a subset of them. We investigate two modes. The first, *Parent-Top1*, retains the child with the higher reward from each parent at the last branching step. This strategy roughly halves gradient computation while ensuring coverage of all parents, yielding stable but slightly less diverse updates. The second, *Extreme selection*, preserves both the globally best and worst $b$ leaves by reward score. This explicitly maintains strong positive and negative signals, which may enhance exploration but also increase variance.

*Depth Pruning.* Branch rollouts generate dense rewards across all denoising steps, but computing gradients at every depth remains costly. To improve efficiency, we introduce **depth pruning**, which skips gradient computation at selected timesteps . Concretely, we maintain a set of pruned depths $\mathcal{D}$ and ignore gradients from nodes at these layers. To prevent permanently discarding certain steps, we adopt a *sliding window* schedule: the pruned depths gradually shift toward later timesteps as training progresses, until reaching a predefined maximum depth. Formally, pruning is active throughout training, and at fixed intervals the window slides one step deeper until the stop depth is reached.

---

**Algorithm 1** BranchGRPO Training Process

---

**Require:** dataset $\mathcal{C}$; policy $\pi_\theta$; behavior policy $\pi_{\theta_{\text{old}}}$; reward models $\{R_k\}$; denoising steps $T$; branching steps $\mathcal{B}$; branching factor $K$; branch correlation $s$

1: **for** iteration $m = 1$ to $M$ **do**
2:     $\pi_{\theta_{\text{old}}} \leftarrow \pi_\theta$
3:     Sample batch $\mathcal{C}_b \subset \mathcal{C}$
4:     **for** prompt $c \in \mathcal{C}_b$ **do**
5:         Sample root noise $z_0 \sim \mathcal{N}(0, I)$
6:         Build rollout tree $\mathcal{T}$ with $\pi_{\theta_{\text{old}}}$:
7:         **for** $t = T$ to $0$ **do**
8:             **if** $t \in \mathcal{B}$ **then**
9:                 Branch into $K$ children with correlation $s$
10:           **else**
11:                Single-step denoising
12:           **end if**
13:         **end for**
14:         Evaluate rewards for leaves $\mathcal{L}(\mathcal{T})$
15:         **Reward fusion:** aggregate leaf rewards upward (path-prob. weights)
16:         **Depth-wise normalization:** standardize per depth, assign edge advantages $A(e)$
17:         **Pruning:** select nodes for backprop only
18:         Form edge set $\mathcal{E}(c)$ from tree $\mathcal{T}$
19:         Compute $J(\theta)$ (clipped-GRPO over $e \in \mathcal{E}(c)$, averaged by $|\mathcal{E}(c)|$)
20:         Update policy: $\theta \leftarrow \theta + \eta \nabla_\theta J(\theta)$
21:     **end for**
22: **end for**

---

## 4 EXPERIMENTS

### 4.1 EXPERIMENT SETUP

We evaluate BranchGRPO on HPDv2.1 (Wu et al., 2023) (103k training and 400 balanced test prompts). The backbone is FLUX.1-Dev (Black Forest Labs, 2024), SD3.5-M Esser et al. (2024) baselines include DanceGRPO and MixGRPO under identical settings. We report efficiency (NFE, iteration time) and quality (HPS-v2.1, PickScore (Kirstain et al., 2023), ImageReward (Xu et al., 2023)), Unified Reward (Wang et al., 2025).

### 4.2 IMPLEMENTATION DETAILS

We set the tree depth to $d = 20$ and the branch factor to $K = 2$, yielding 16 leaves per rollout before pruning. The branching steps $\mathcal{B}$ use three presets: Dense $(0, 3, 6, 9)$ as the default, Mixed $(0, 4, 8, 12)$, and Sparse $(0, 5, 10, 15)$. The branch correlation sweeps $s \in \{1, 2, 4, 8\}$. Training runs for 300 optimizer steps with gradient accumulation $g = 12$ and per-GPU batch size $= 2$, on $16\times$ NVIDIA H200 GPUs. Optimization uses AdamW (learning rate $1 \times 10^{-5}$, weight decay $1 \times 10^{-4}$) with bf16 precision and EMA weights stored on CPU. All GRPO-related hyperparameters are kept identical across methods, with full details deferred to the supplementary material.

### 4.3 MAIN RESULTS

Table 1 summarizes efficiency and alignment performance. BranchGRPO consistently outperforms baselines across human-preference metrics while offering favorable compute trade-offs. In particular, BranchGRPO-DepthPruning achieves the best overall alignment, raising HPS-v2.1 from 0.360 (DanceGRPO) to 0.369 and delivering the highest PickScore (0.231), ImageReward (1.625), and Unified Reward (3.404). BranchGRPO-WidthPruning and BranchGRPO-Mix further reduce iteration time to 314s and 148s, respectively, with only marginal drops in quality—making them highly practical for large-scale training. Compared with MixGRPO (289s, HPS-v2.1=0.359, Unified Reward=3.380), BranchGRPO variants yield both stronger alignment and more flexible scaling.

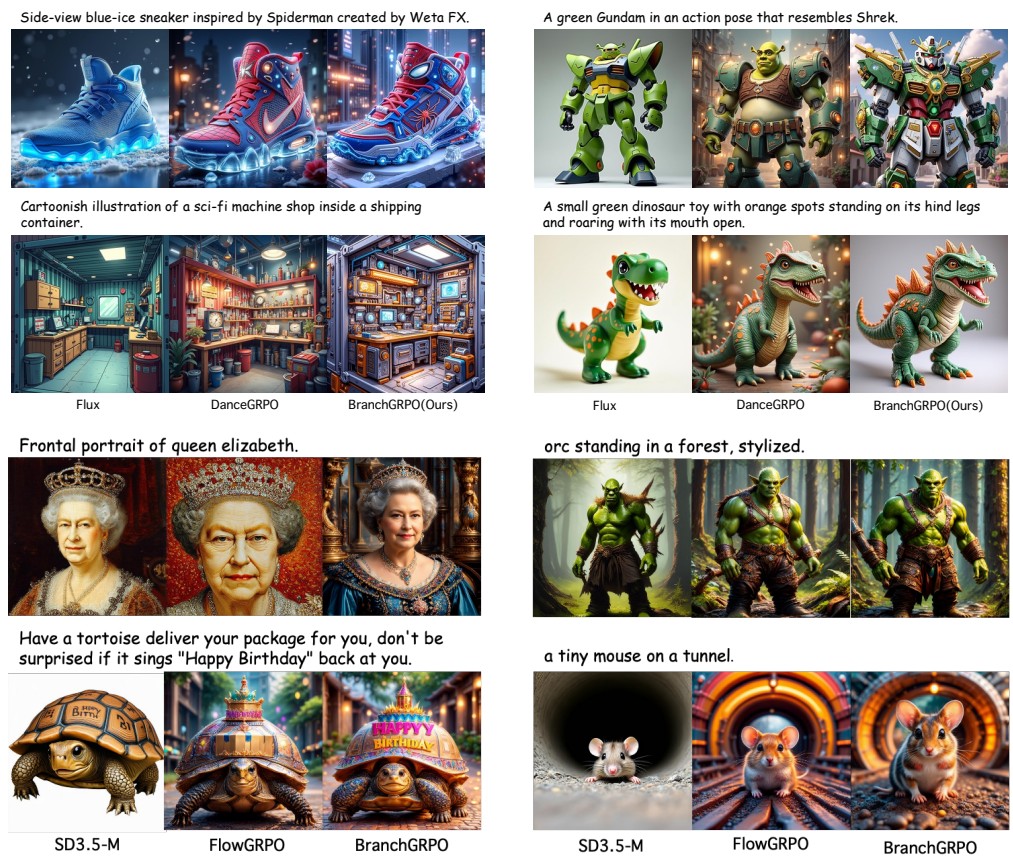

Figure 4: Qualitative comparison of generations. *Top:* Flux, DanceGRPO, and our BranchGRPO. *Bottom:* SD3.5-M, FlowGRPO, and our BranchGRPO.

Reward curves in Figure 1 confirm these findings: DanceGRPO(tf=0.6) suffers from instability, while the full-timestep variant converges more smoothly but at high cost. BranchGRPO achieves faster early reward growth, smoother convergence, and higher final rewards. Qualitative comparisons in Figure 4 further show that our method produces sharper details and better semantic alignment than Flux and DanceGRPO.

Table 2 further validates the generality of BranchGRPO on the SD3.5-M backbone. When plugged into FlowGRPO, BranchGRPO reduces the total GPU-hours (from 2000 to 1460) while keeping the same number of training steps. Despite using *less than half* the compute, BranchGRPO consistently improves all alignment metrics—HPS-v2.1, PickScore, ImageReward, and GenEval. These results show that BranchGRPO transfers cleanly across different backbones and GRPO-style pipelines, providing both higher training efficiency and stronger alignment quality under the same—or even lower—compute budgets.

**Reward–KL Efficiency.** Beyond final reward and training speed, we also examine the reward–KL tradeoff, which measures how efficiently an RL policy converts KL divergence into reward gains. As shown on the right, BranchGRPO-DepthPruning consistently lies above the DanceGRPO frontier across the full KL range. This indicates that BranchGRPO extracts more reward per unit KL, reflecting more stable credit assignment and better optimization efficiency.

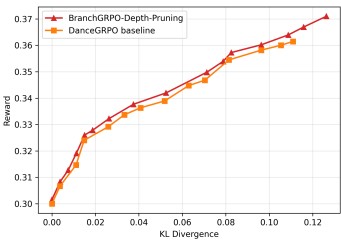

Figure 5: Reward–KL curves comparing BranchGRPO-DepthPruning and DanceGRPO.

Table 1: Efficiency–quality comparison. The best and second-best results in each column are highlighted in **bold** and underline, respectively. NFE denotes the number of function evaluations of the denoiser. For branching methods, we report the *average per-sample NFE*, computed as the total function evaluations in the tree divided by the number of final samples.

| Method | $NFE_{\pi_{\theta_{old}}}$ | $NFE_{\pi_\theta}$ | Iteration Time (s)↓ | HPS-v2.1↑ | Pick Score↑ | Image Reward↑ | Unified Reward↑ |
|---|---|---|---|---|---|---|---|
| FLUX | - | - | - | 0.313 | 0.227 | 1.112 | 3.370 |
| DanceGRPO(tf=1.0) | 20 | 20 | 698 | 0.360 | 0.234 | 1.612 | 3.388 |
| DanceGRPO(tf=0.6) | 20 | 12 | 469 | 0.353 | 0.228 | 1.517 | 3.362 |
| MixGRPO (20,5) | 20 | 5 | 289 | 0.359 | 0.228 | 1.594 | 3.380 |
| BranchGRPO | 13.68 | 13.68 | 493 | 0.363 | 0.229 | 1.603 | 3.386 |
| BranchGRPO-WidPru | 13.68 | 8.625 | 314 | 0.364 | 0.231 | 1.609 | 3.383 |
| BranchGRPO-DepPru | 13.68 | 8.625 | 314 | **0.369** | **0.235** | **1.625** | **3.404** |
| BranchGRPO-Mix | 13.68 | 4.25 | **148** | 0.363 | 0.230 | 1.598 | 3.384 |

Table 2: Generalization on SD3.5-M and integration into GRPO-style training pipelines. BranchGRPO consistently improves alignment quality and training efficiency.

| Method | GPU Hours↓ | HPS-v2.1↑ | Pick Score↑ | Image Reward↑ | GenEval↑ |
|---|---|---|---|---|---|
| SD3.5-M | – | 0.204 | 20.51 | 0.85 | 0.63 |
| FlowGRPO | 2000 | 0.316 | 23.50 | 1.29 | 0.86 |
| FlowGRPO | **1000** | 0.280 | 22.41 | 0.95 | 0.73 |
| BranchGRPO | 1460 | **0.323** | **23.58** | **1.32** | **0.89** |

## 4.4 ABLATION STUDIES

We conduct a series of ablation studies to better understand the design choices in BranchGRPO. Unless otherwise stated, all are carried out under the same training setup as in Section 4.3. The following analyses highlight how different branching configurations and aggregation strategies affect efficiency, reward quality, and stability.

*Branch Correlation.* Figure 6(a) shows the effect of varying the branch correlation $s$. Smaller values ($s = 1.0, 2.0$) limit exploration and lead to slower reward growth, while very large values ($s = 8.0$) destabilize early training. A moderate setting ($s = 4.0$) achieves the best trade-off, reaching the highest reward and stable convergence, confirming that stochastic branching is necessary but should be carefully tuned.

*Branching Steps.* We next vary the positions of split timesteps (Figure 6(b)). Early splits such as $(0, 3, 6, 9)$ promote faster reward increase in the early stage, whereas later splits like $(9, 12, 15, 18)$ delay exploration and yield lower rewards. Intermediate schedules (e.g., $(3, 6, 9, 12)$) balance efficiency and reward quality, suggesting that early splits are generally more effective for exploration.

*Branch Density.* Finally, we compare different densities of split points while keeping the overall horizon fixed (Figure 6(c)). Although all configurations eventually converge to similar reward levels, denser splits (e.g., $(0, 3, 6, 9)$) accelerate early training, while sparser configurations converge more slowly. This indicates that increasing the density of branching in the early phase improves sample efficiency without harming stability.

*Branch Rollout Diversity.* A potential concern is that Branch Rollout may reduce diversity. Beyond the distribution-level checks in Sec. 3.1, we also evaluate prompt-conditioned diversity using LPIPS-MPD and TCE following prior work Kim et al. (2025). As shown on the right, BranchGRPO remains very close to DanceGRPO across branching schedules, indicating that sample diversity is essentially unchanged.

Table 3: Prompt-conditioned diversity under different branching schedules (new).

| Method | LPIPS-MPD ↑ | TCE ↑ |
|---|---|---|
| DanceGRPO | 0.723 | 4.45 |
| BranchGRPO (0,2,4,8) | 0.719 | 4.44 |
| BranchGRPO (0,3,6,9) | 0.713 | 4.42 |
| BranchGRPO (0,4,8,12) | 0.704 | 4.39 |

*Reward Fusion Strategies.* Figure 6(d) compares uniform averaging ($\beta = 0$) with path-probability weighting ($\beta = 1$) under identical training settings. Uniform averaging shows higher variance and a

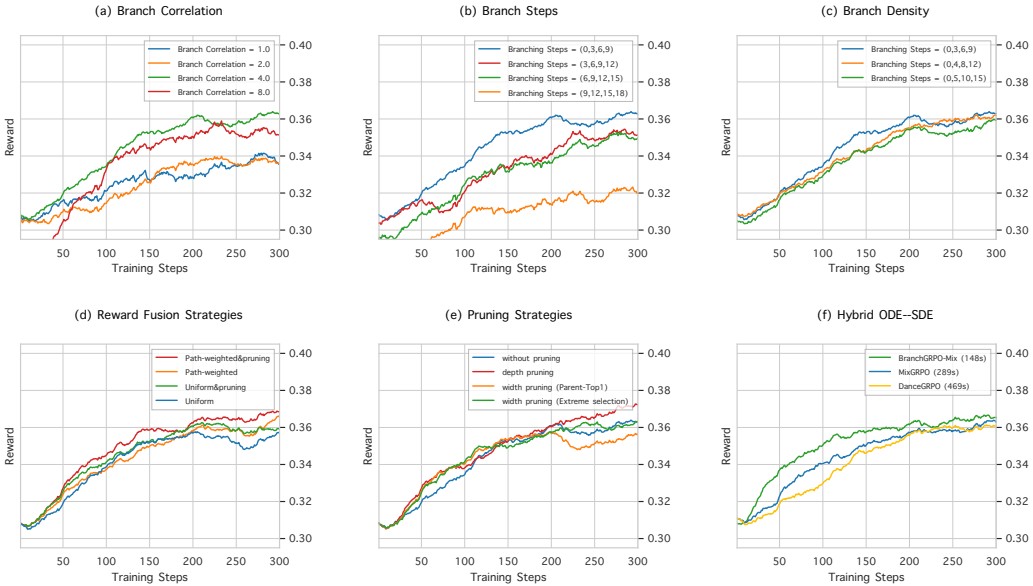

Figure 6: Ablation studies of BranchGRPO. Moderate branch correlation, early and denser splits improve reward growth; path-weighted fusion enhances stability; depth pruning achieves the best final reward; and the hybrid ODE–SDE provides the fastest training speed while remaining stable.

clear late-stage plateau, whereas path-weighted fusion delivers consistently higher and more stable rewards throughout training. This empirically supports Sec. 3.3: uniform averaging is exploration-friendly but noisy, while path weighting aligns credit assignment with the behavior distribution and improves convergence at no extra cost.

*Pruning Strategies.* Figure 6(e) compares pruning methods applied *after* depth-wise normalization and *only during backpropagation*. For *depth pruning*, we adopt a sliding-window schedule over denoising steps, the window is initialized at the last split point, has a fixed size of $4$, and shifts by one denoising step every $30$ training iterations. GRPO losses and gradients in the active window are skipped, while forward sampling remains unchanged. This schedule yields the **best final reward** and reveals substantial redundancy at late timesteps. For *width pruning (Parent-Top1)*, we retain only the locally better child at each branch for gradient updates, effectively halving updates and producing the smoothest, lowest-variance curve, though with slightly lower final reward than depth pruning. *Width pruning (Extreme-b)* keeps both the globally best and worst $b$ leaves, injecting stronger positive/negative signals and remaining competitive at the end, but with higher variance.

*Hybrid ODE–SDE.* To further explore depth pruning, inspired by MixGRPO (Li et al., 2025a), we design a *hybrid ODE–SDE schedule*: all branching steps are preserved as SDE, while a sliding window determines additional SDE steps, with the remaining updates replaced by ODE. Figure 6(f) shows that this scheme achieves the fastest speedups (148s vs. 289s for MixGRPO vs. 469s for DanceGRPO ) while maintaining stable and fast reward growth.

## 4.5 SCALING WITH BRANCH

DanceGRPO scales poorly: one GRPO training step with 81 rollout samples takes over 3500s, whereas BranchGRPO achieves the same scale in only 680s, making large-scale scaling feasible. We investigate two settings: scaling the branch factor ($K{=}2, 3, 4$ yielding 16, 81, and 256 leaves) and scaling the number of branching steps ($3, 4, 5$ splits yielding 8, 16, and 32 leaves under $K{=}2$).

As shown in Figure 7, scaling along both dimensions leads to clear and substantial gains in reward growth and final performance. Larger branch factors and more branching steps consistently push the reward curves higher, with improvements becoming increasingly pronounced as the rollout tree expands. This demonstrates that BranchGRPO can effectively generate additional samples, making large-scale scaling both practical and beneficial without compromising stability.

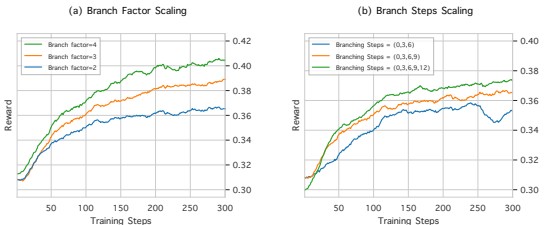

Figure 7: Impact of scaling branch rollouts in BranchGRPO. Larger branch factors (a) and more branching steps (b) consistently improve reward, following a clear scaling law.

| Method | Group Size | GPU Hours↓ | Reward↑ |
|--------|-----------|-----------|---------|
| DanceGRPO | 16 | 928 | 0.360 |
| BranchGRPO | 16 | **192** | 0.363 |
| BranchGRPO | 32 | 368 | 0.373 |
| BranchGRPO | 64 | 787 | 0.381 |
| BranchGRPO | 81 | 906 | 0.387 |
| BranchGRPO | 256 | 3072 | **0.404** |

Table 4: Here we use BranchGRPO-Mix. Under similar GPU-hours, BranchGRPO with group size 81 achieves a reward of 0.387, substantially outperforming Dance-GRPO (0.360).

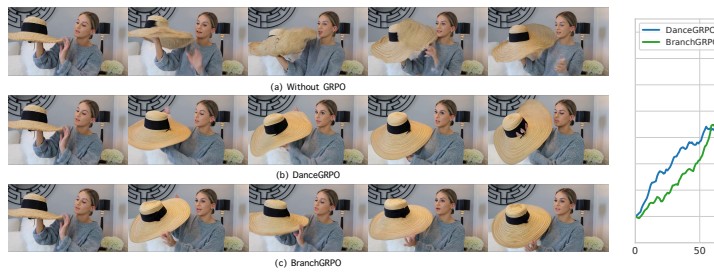

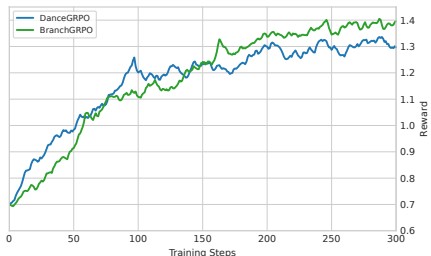

Figure 8: Video generation results on Wan2.1-1.3B. Left: qualitative frame comparisons across three settings. Right: reward curves showing faster convergence and higher final rewards with BranchGRPO compared to DanceGRPO.

## 4.6 VIDEO GENERATION RESULTS

We further evaluate BranchGRPO on video generation using Wan2.1-1.3B (Wan et al., 2025), with the Video-Align's motion quality (Liu et al., 2025b) as reward.

As shown in Figure 8, the base model exhibits severe temporal flickering and deformation, while DanceGRPO improves consistency but still produces blurry details. BranchGRPO generates sharper and more coherent frames across time, and the reward curves demonstrate faster convergence and higher final rewards compared to DanceGRPO.

These results highlight that branching rollouts are particularly effective for video generation, where reward sparsity and temporal coherence are especially challenging. In practice, BranchGRPO also improves efficiency: each iteration takes only about 8 minutes, compared to 20 minutes for Dance-GRPO. Additional visual examples are provided in the supplementary material.

## 5 CONCLUSION

We introduced **BranchGRPO**, a tree-structured GRPO paradigm that replaces sequential rollouts with prefix-sharing branching and depth-wise reward fusion, augmented by lightweight pruning for compute reallocation. Across image and video generation, BranchGRPO yields faster convergence, more stable training, and higher final alignment quality under matched budgets. These results establish structured branching as a practical, scalable path for RLHF in diffusion and flow models, and we further verify a scaling trend in which larger group sizes consistently improve performance.

ACKNOWLEDGMENTS

This work was supported by the National Natural Science Foundation of China (No. 62476011 and No. 62576043), and by the Beijing Natural Science Foundation (No. L252060).

ETHICS STATEMENT

This work focuses on improving optimization efficiency and stability for diffusion/flow-based generative models. Our experiments rely on publicly available prompt sets and reward models cited in the paper. No personally identifiable information, human subjects data, or sensitive attributes were collected or annotated by the authors. We followed the licenses and usage terms of all datasets and third-party models; when applicable, we restricted use to non-commercial research. Given that generative models may amplify biases in reward models or datasets, we report failure cases and ablations that probe stability and variance, and we avoid deploying the models in safety-critical settings. We will release code and configuration files to facilitate scrutiny and responsible reuse.

REPRODUCIBILITY STATEMENT

We release anonymized training and evaluation code, configuration files, and exact hyperparameters in the supplementary materials. All experiments can be reproduced using the provided scripts, seeds, and environment specifications.

LLM USAGE

An LLM was used *only for language polishing* of the manuscript (e.g., grammar, wording, and minor clarity edits). No experimental results were generated, modified, or selected by the LLM; all technical content, claims, ideas, and analyses are the authors' own. No confidential or non-public data were shared with the LLM. The authors take full responsibility for the final content.

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

APPENDIX

This appendix provides additional theoretical proofs, implementation details, and experimental results that complement the main paper. The contents are organized as follows:

- **Section A**: Hyperparameter settings used in all experiments.
- **Section B**: Theoretical analysis, including proofs for branch noise construction, reward fusion, and variance reduction.
- **Section C**: Additional experiments, including ablations, more text-to-image and image-to-video results. **Failure Cases**: Qualitative examples where BranchGRPO fails, highlighting current limitations.
- **Section D**: Discussion and future work.

## A  HYPERPARAMETER SETTINGS

Table 5 summarizes the detailed hyperparameter configuration used in our experiments. All hyperparameters are kept identical across all methods, including **DanceGRPO** and **MixGRPO**, to ensure a fair comparison. For depth pruning and hybrid-ODE-SDE, we follow the design of **MixGRPO** and adopt a sliding window of size 4, which shifts one step deeper every 30 iterations.

Table 5: Hyperparameter settings used in all experiments.

| Parameter | Value | Parameter | Value |
|---|---|---|---|
| Random seed | 42 | Learning rate | $1 \times 10^{-5}$ |
| Train batch size | 2 | Weight decay | $1 \times 10^{-4}$ |
| SP size | 1 | Mixed precision | bfloat16 |
| SP batch size | 2 | Grad. checkpointing | Enabled |
| Dataloader workers | 4 | Max grad norm | 0.01 |
| Grad. accum. steps | 12 | Warmup steps | 0 |
| Checkpoint steps | 40 | Use TF32 | Yes |
| Resolution | $720 \times 720$ | Sampling steps | 16 |
| Eta | 0.3 | Sampler seed | 1223627 |
| Num. generations | 12 | Shift (branch offset) | 3 |
| Use group reward | Yes | Ignore last step | Yes |
| Clip range | $1 \times 10^{-4}$ | Adv. clip max | 5.0 |
| Use EMA | Yes | EMA decay | 0.995 |
| Init same noise | Yes | | |

Table 6: **Recommended default hyperparameter settings for BranchGRPO.** These values form a unified and robust configuration that works across all tasks (image/video) and backbones (Flux, SD3.5-M).

| Component | Parameter | Default | Stable Range |
|---|---|---|---|
| Branching | Branch points | (0,3,6,9) | |
| | Branching factor | 2 | (2,3,4) |
| Noise Correlation | Correlation scale ($s$) | 4.0 | (3.0-5.0) |
| Reward Fusion | Temperature ($\beta$) | 0 | 0 or 1 |
| Pruning | Pruning type | depth pruning | depth or width pruning |
| | Depth window size | 5 | () |
| Hybrid ODE–SDE | Mix ratio | 30% | |

To further guide practitioners, we identify two optimal configurations based on resource priorities:

- **Scenario 1: Maximum Performance (Scaling Mode).**
  If the goal is to push the upper bound of alignment performance, we recommend using **BranchGRPO-Mix** and scaling the group size to $N = 81$ (e.g., Branch Factor $K = 3$, Number of Branching Steps = 4).
    - **Benefit:** Due to the efficiency of BranchGRPO-Mix, this configuration maintains an iteration time and total GPU-hour budget similar to standard DanceGRPO/FlowGRPO (at $N = 16$).
    - **Outcome:** Under this similar compute budget, it yields **significant performance gains** (HPS $0.360 \rightarrow 0.387$) by leveraging massive exploration that is computationally prohibitive for sequential baselines.
- **Scenario 2: Balanced Efficiency & Quality (Universal Default).**
  If the goal is to accelerate training while maintaining or slightly improving quality, we recommend **BranchGRPO-DepthPruning** with the default settings in Table 6.
    - **Benefit:** It delivers a **2.2× speedup** (314s vs. 698s per iteration) compared to the baseline.
    - **Outcome:** It achieves strictly better alignment (HPS $0.360 \rightarrow 0.369$) without harming the original model's diversity or capacity.

## B  THEORETICAL ANALYSIS

### B.1  BRANCH NOISE CONSTRUCTION AND BOUNDARY DISTRIBUTION PRESERVATION

We use a reverse-time grid $t_0 > t_1 > \cdots > t_N$ so that $h_i = t_i - t_{i+1} > 0$.

Consider the reverse SDE discretized by Euler–Maruyama:

$$z_{i+1} = \mu_\theta(z_i, t_i) + g_i\, \eta_i, \qquad \eta_i \sim \mathcal{N}(0, I), \;\; g_i := g(t_i)\sqrt{h_i}. \tag{6}$$

At a split step $i \in \mathcal{B}$, we construct $K$ branch noises as

$$\xi_b = \frac{\xi_0 + s\,\eta_b}{\sqrt{1 + s^2}}, \qquad \xi_0, \eta_b \overset{i.i.d.}{\sim} \mathcal{N}(0, I), \tag{7}$$

with fresh $(\xi_0, \{\eta_b\})$ drawn at each split step and no cross-time sharing. Then $\xi_b \sim \mathcal{N}(0, I)$ for each $b$, and $\mathrm{Cov}(\xi_b, \xi_{b'}) = \frac{1}{1+s^2} I$ for $b \neq b'$.

Each child branch then updates as

$$z_{i+1}^{(b)} = \mu_\theta(z_i, t_i) + g_i\, \xi_b. \tag{8}$$

**Lemma 1** (Single-step marginal preservation). *For any fixed parent $z_i$, we have*

$$z_{i+1}^{(b)} \overset{d}{=} \mu_\theta(z_i, t_i) + g_i\eta_i, \qquad \eta_i \sim \mathcal{N}(0, I).$$

**Lemma 2** (Leaf marginal preservation). *Assuming independent noises across time steps and no cross-time reuse of the shared component, conditioned on prefix $(z_0, \eta_0, \ldots, \eta_{i-1})$, each branch $z_N^{(b)}$ generated by the split rule has the same distribution as a baseline SDE sample $z_N$.*

**Theorem 1** (Boundary distribution invariance). *Under the branching construction above, for any set of split steps $\mathcal{B}$, each leaf $z_N^{(b)}$ has the same marginal law as a baseline SDE rollout. Hence, branching does not alter the final generator distribution.*

### B.2  REWARD FUSION: UNBIASEDNESS AND VARIANCE REDUCTION

Let $L(n)$ be the leaf set of a node $n$. Each leaf has reward $r_\ell = r(z_N^{(\ell)})$. Define the conditional expected return

$$V(n) := \mathbb{E}[\, r(z_N) \mid n \,].$$

**Uniform fusion.**

$$\bar{r}(n) = \frac{1}{|L(n)|} \sum_{\ell \in L(n)} r_\ell. \tag{9}$$

Then

$$\mathbb{E}[\bar{r}(n) \mid n] = V(n), \qquad \mathrm{Var}(\bar{r}(n) \mid n) = \frac{\sigma^2(n)}{|L(n)|},$$

where $\sigma^2(n) = \mathrm{Var}(r_\ell \mid n)$.

**Path-probability weighted fusion.**  If leaves are drawn from proposal $q(\ell \mid n)$ with weights

$$w_\ell = \frac{p_{\text{beh}}(\ell \mid n)}{q(\ell \mid n)},$$

then the IS estimator

$$\hat{r}_{\text{IS}}(n) = \frac{1}{|L(n)|} \sum_{\ell \in L(n)} w_\ell r_\ell$$

is unbiased: $\mathbb{E}[\hat{r}_{\text{IS}}(n) \mid n] = V(n)$. The self-normalized IS estimator

$$\hat{r}_{\text{SNIS}}(n) = \frac{\sum_\ell w_\ell r_\ell}{\sum_\ell w_\ell}$$

is consistent with variance $O(1/\text{ESS})$, where

$$\text{ESS} = \frac{(\sum_\ell w_\ell)^2}{\sum_\ell w_\ell^2}.$$

*Remark.* In practice (Eq. (4) in the main text), we adopt a softmax weighting $w_\ell \propto \exp(\beta s_\ell)$ based on path log-probabilities $s_\ell$. This can be interpreted as a temperature-smoothed variant of SNIS, where $\beta$ controls the sharpness of importance weights. Although no longer strictly unbiased, this form provides stable training and reduces the influence of low-probability noisy leaves.

### B.3 DEPTH-WISE BASELINE (CONTROL VARIATES)

For $K$ siblings at depth $i$, let fused returns $\bar{r}^{(b)}$, and group mean $\bar{r}_i = \frac{1}{K}\sum_b \bar{r}^{(b)}$. Define

$$A_i^{(b)} = \bar{r}^{(b)} - \bar{r}_i, \qquad \sum_b A_i^{(b)} = 0. \tag{10}$$

Let $g_i^{(b)}(\theta) = \nabla_\theta \log p_\theta(\text{branch } b \text{ at depth } i)$. Then

$$\widehat{\nabla J}_{\text{group}} = \frac{1}{K}\sum_{b=1}^{K} A_i^{(b)} g_i^{(b)}$$

is an unbiased gradient estimator with strictly smaller variance than $\widehat{\nabla J}_{\text{single}} = \frac{1}{K}\sum_b \bar{r}^{(b)} g_i^{(b)}$, unless $\text{Cov}(\bar{r}^{(b)}, g_i^{(b)}) = 0$.

### B.4 CONTINUOUS REWARDS AND CONCENTRATION

Assume $r$ is $L$-Lipschitz and the SDE flow $\Psi_{i+1 \to N}$ is $K_i$-Lipschitz. Then

$$|r(z_N^{(b)}) - r(z_N^{(b')})| \leq LK_i g_i \|\xi_b - \xi_{b'}\|.$$

Thus $r(z_N^{(b)})$ is sub-Gaussian with parameter $\mathcal{O}(L^2 K_i^2 g_i^2)$. Averaging over $|L(n)|$ leaves gives

$$\Pr\left(|\bar{r}(n) - V(n)| \geq \varepsilon \mid n\right) \leq 2\exp\left(-c \cdot |L(n)|\,\varepsilon^2/(L^2 K_i^2 g_i^2)\right).$$

## C ADDITIONAL EXPERIMENTS

### C.1 MORE QUANTITATIVE RESULT

Table 7: Effect of applying BranchGRPO *only* to the rollout procedure in DiffusionNFT.

| Method | Iter. Time(s)↓ | GPU Hours↓ | HPS-v2.1↑ | PickScore↑ |
|---|---|---|---|---|
| DiffusionNFT | 159 | 706 | 0.331 | 23.80 |
| DiffusionNFT + BranchGRPO (rollout only) | **112** | 497 | 0.328 | 23.74 |

Table 8 reports more results.

we conducted a small human A/B study (100 prompts, 5 annotators). As shown in Table **??**, BranchGRPO is preferred over DanceGRPO in **47%** of comparisons, consistent with its higher HPSv2.1 score. This verifies that the observed improvements are reflected in actual human preference rather than reward-model.

Table 8: Ablation study of BranchGRPO under different design choices. Best and second-best per column are in **bold** and underline. All results are obtained under the same training setup as in Section 4.3.

| Configuration | NFE$_{\pi_{\theta_{old}}}$ | NFE$_{\pi_\theta}$ | Iteration Time (s)↓ | HPS-v2.1↑ | Pick Score↑ | Image Reward↑ | CLIP Score↑ |
|---|---|---|---|---|---|---|---|
| *Branch Density* | | | | | | | |
| (0, 3, 6, 9) | 13.68 | 13.68 | 493 | 0.363 | 0.229 | 1.603 | 0.374 |
| (0, 4, 8, 12) | 11.56 | 11.56 | 416 | 0.359 | 0.229 | 1.594 | 0.374 |
| (0, 5, 10, 15) | 9.44 | 9.44 | 340 | 0.354 | 0.228 | 1.565 | 0.366 |
| *Reward Fusion* | | | | | | | |
| Uniform Fusion | 13.68 | 13.68 | 493 | 0.361 | 0.229 | 1.595 | 0.368 |
| Path-Weighted Fusion | 13.68 | 13.68 | 493 | 0.363 | 0.229 | 1.603 | 0.374 |
| *Pruning Strategy* | | | | | | | |
| Width Pruning | 13.68 | 8.625 | 314 | 0.364 | 0.231 | 1.609 | 0.374 |
| Depth Pruning | 13.68 | 8.625 | **314** | **0.369** | **0.235** | **1.625** | **0.381** |

Table 9: **Human preference and Best-of-N scaling results.**

(a) Human preference evaluation on 100 prompts with 5 annotators.

| Method | Flux | DanceGRPO | BranchGRPO |
|---|---|---|---|
| User Study | 19% | 33% | 48% |

(b) Best-of-N (BoN) Reward Scaling on HPS-v2.1.

| Method | N=1 | N=2 | N=4 | N=8 | N=16 |
|---|---|---|---|---|---|
| DanceGRPO | 0.3524 | 0.3581 | 0.3626 | 0.3663 | 0.3670 |
| **BranchGRPO** | **0.3613** | **0.3666** | **0.3706** | **0.3737** | **0.3742** |

Table 10: GenEval Result.

| Model | Overall ↑ | Single Obj. ↑ | Two Obj. ↑ | Counting ↑ | Colors ↑ | Position ↑ | GPU Hours ↓ |
|---|---|---|---|---|---|---|---|
| **SD3.5-M** | 0.63 | 0.98 | 0.78 | 0.50 | 0.81 | 0.24 | - |
| **FlowGRPO** | 0.86 | 0.99 | 0.95 | 0.89 | 0.88 | 0.83 | 2000 |
| **FlowGRPO** | 0.73 | 0.99 | 0.85 | 0.77 | 0.81 | 0.67 | 1000 |
| **BranchGRPO** | 0.89 | 0.99 | 0.96 | 0.90 | 0.89 | 0.85 | 1460 |

Table 11: **BranchGRPO on Qwen-Image.** We report GPU-hours, per-iteration time, and four alignment metrics.

| Method | GPU Hours ↓ | Iter. Time (s) ↓ | HPS-v2.1 ↑ | PickScore ↑ | ImageReward ↑ | GenEval ↑ |
|---|---|---|---|---|---|---|
| Qwen-Image (Base) | — | — | 0.263 | 22.21 | 0.87 | 0.87 |
| +DanceGRPO | 389 | 584 | 0.318 | 23.03 | 1.13 | 0.89 |
| +BranchGRPO-DepPru | **184** | **276** | **0.321** | **23.17** | **1.19** | **0.90** |

Table 12: CLIPScore analysis under HPS-only and multi-objective training. All HPS-only GRPO variants reduce prompt adherence; a simple multi-objective setup partially recovers CLIPScore while maintaining strong HPS.

| Method | Reward Model | HPSv2 (↑) | CLIP Score (↑) |
|---|---|---|---|
| Flux (Base Model) | - | 0.313 | **0.405** |
| DanceGRPO(tf=1.0) | HPS-v2.1 only | 0.360 | 0.371 |
| DanceGRPO(tf=0.6) | HPS-v2.1 only | 0.353 | 0.375 |
| BranchGRPO(0,3,6,9) | HPS-v2.1 only | 0.363 | 0.369 |
| BranchGRPO(0,4,8,12) | HPS-v2.1 only | 0.359 | 0.371 |
| BranchGRPO(0,5,10,15) | HPS-v2.1 only | 0.354 | 0.376 |
| **DanceGRPO**(tf=0.6) | HPS-v2.1 + CLIP Score | 0.334 | 0.398 |
| **BranchGRPO(0,3,6,9)** | HPS-v2.1 + CLIP Score | 0.339 | 0.392 |

## C.2 MORE TEXT2IMAGE RESULTS

An anime man in flight uniform with hyper detailed digital artwork and an art style inspired by Klimt, Nixeu, Ian Sprigger, Wlop, and Krenz Cushart.

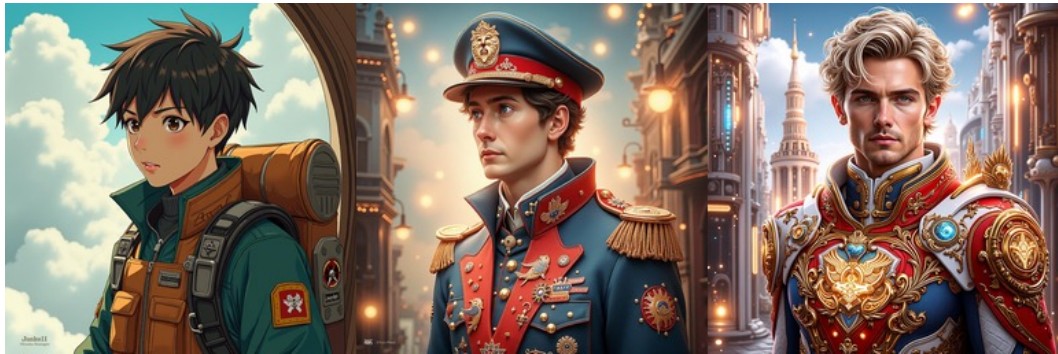

|  Flux | DanceGRPO | BranchGRPO(Ours) |

A raccoon riding an oversized fox through a forest in a furry art anime still.

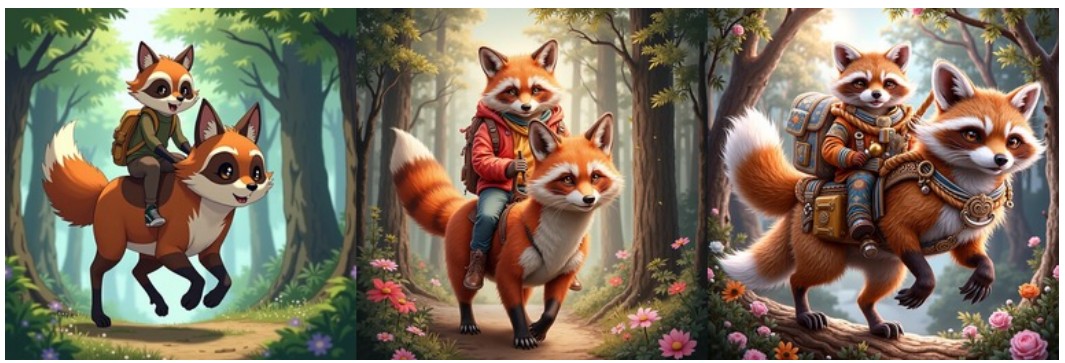

|  Flux | DanceGRPO | BranchGRPO(Ours) |

Totem pole made out of cats.

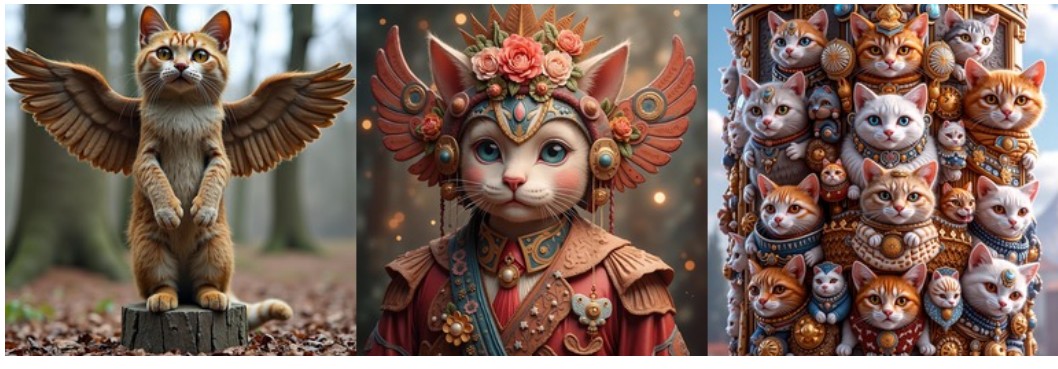

|  Flux | DanceGRPO | BranchGRPO(Ours) |

Figure 9

A portrait of two women with purple hair flying in different directions against a dark background.

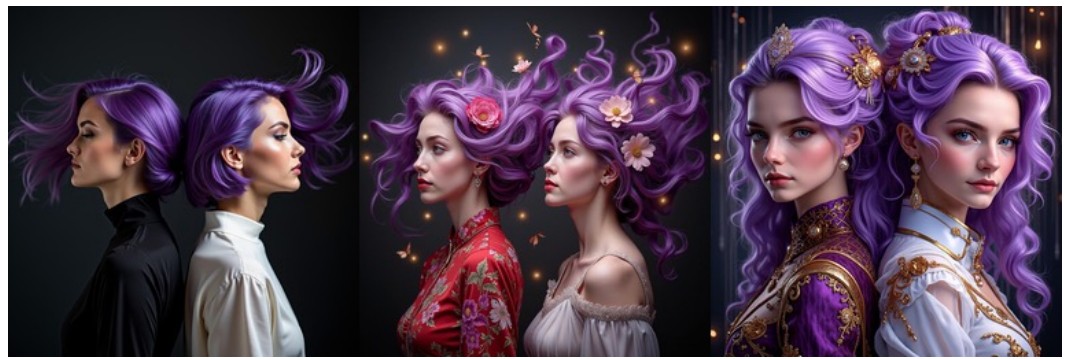

Flux                    DanceGRPO                    BranchGRPO(Ours)

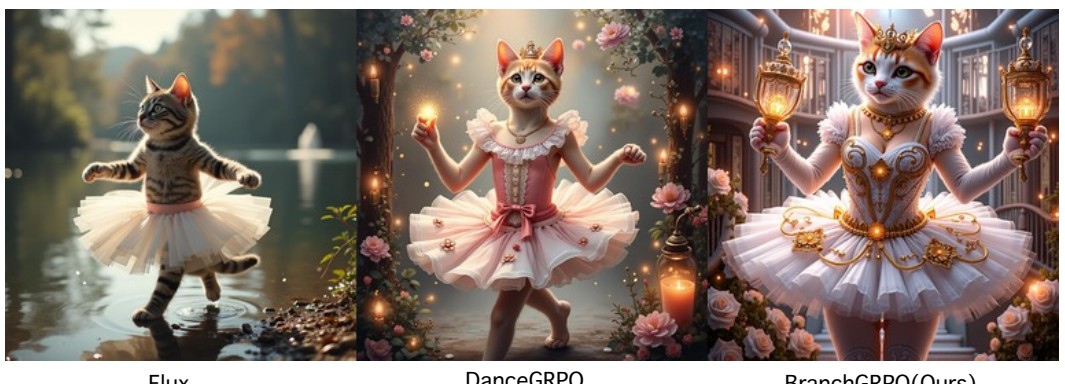

Flux                    DanceGRPO                    BranchGRPO(Ours)

A cute anime schoolgirl with a sad face submerged in dark pink and blue water, portrayed in an oil painting style.

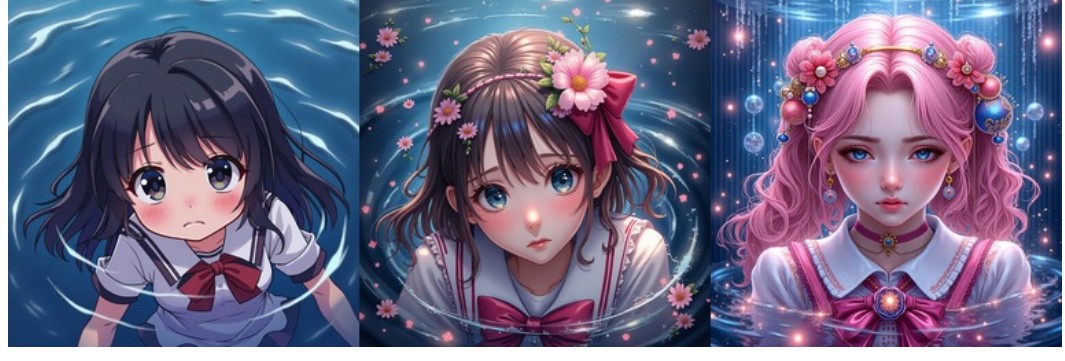

Flux                    DanceGRPO                    BranchGRPO(Ours)

Figure 10

The image is of a raccoon wearing a Peaky Blinders hat, surrounded by swirling mist and rendered with fine detail.

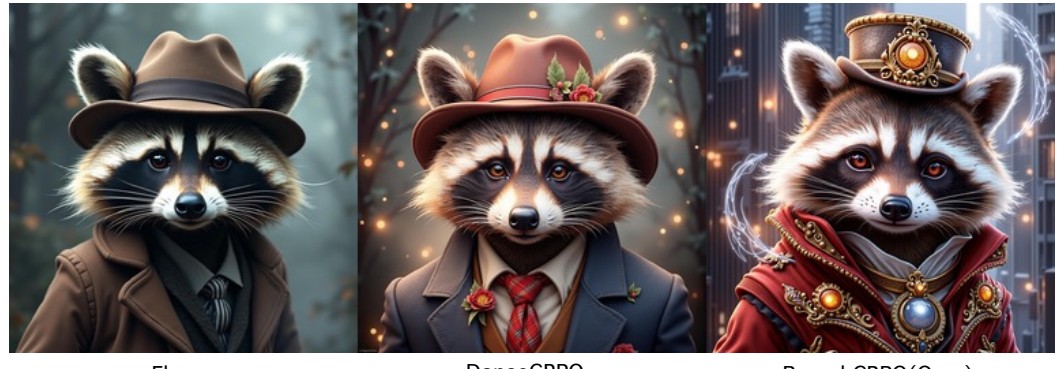

Flux                              DanceGRPO                         BranchGRPO(Ours)

Portrait of an anime princess in white and golden clothes.

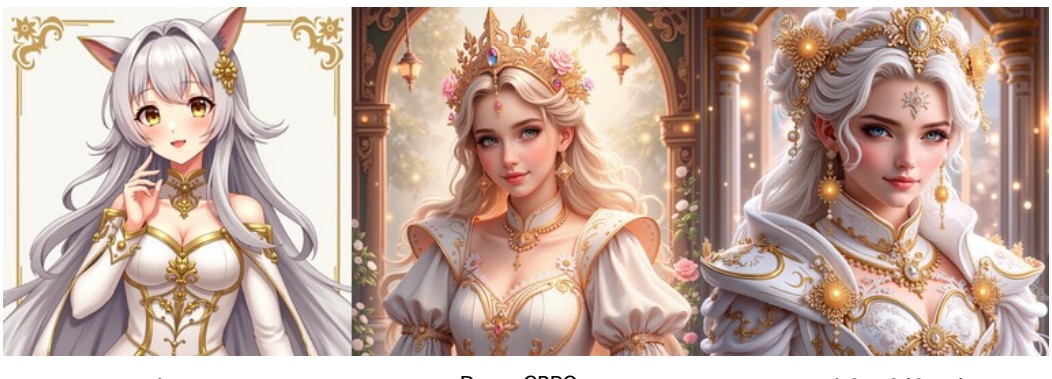

Flux                              DanceGRPO                         BranchGRPO(Ours)

A cute little anthropomorphic Tropical fish knight wearing a cape and a crown in short, pale blue armor.

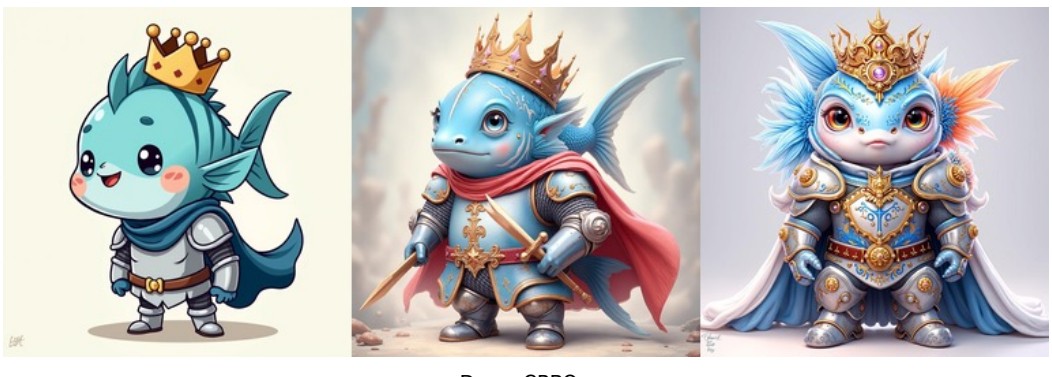

Flux                              DanceGRPO                         BranchGRPO(Ours)

Figure 11

A white polar bear cub wearing sunglasses sits in a meadow with flowers.

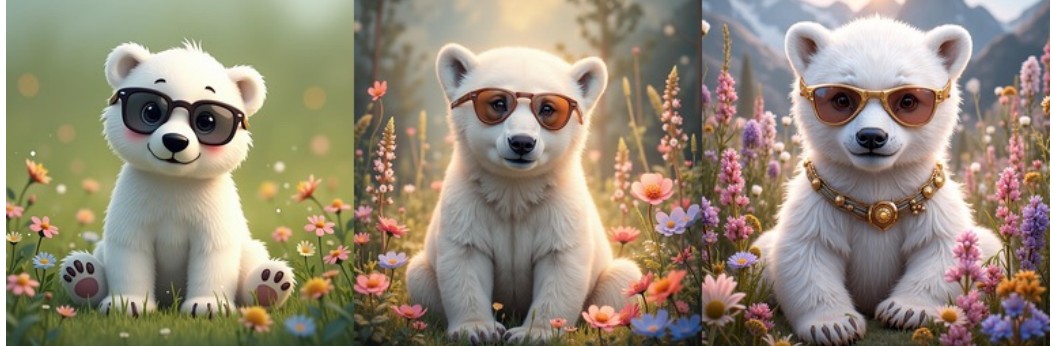

Flux DanceGRPO BranchGRPO(Ours)

A photo of a mechanical angel woman with crystal wings, in the sci-fi style of Stefan Kostic, created by Stanley Lau and Artgerm.

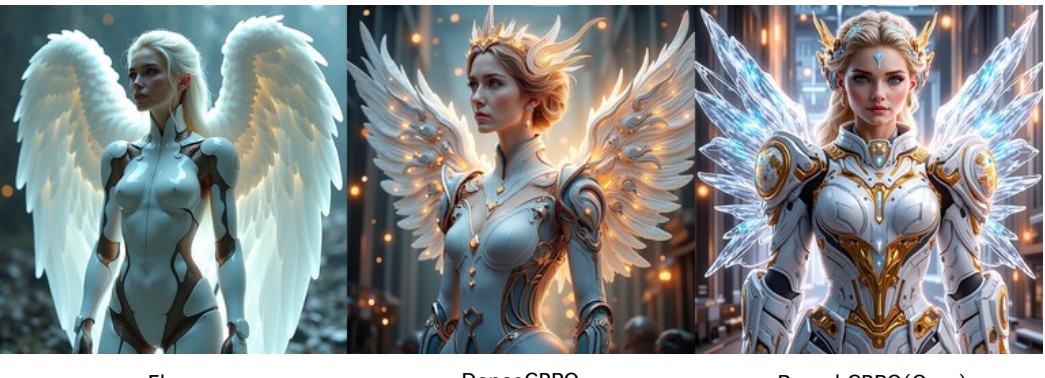

Flux DanceGRPO BranchGRPO(Ours)

Close-up shot of a person running on a treadmill with worn running shoes under dramatic lighting and a comic book-style painting effect.

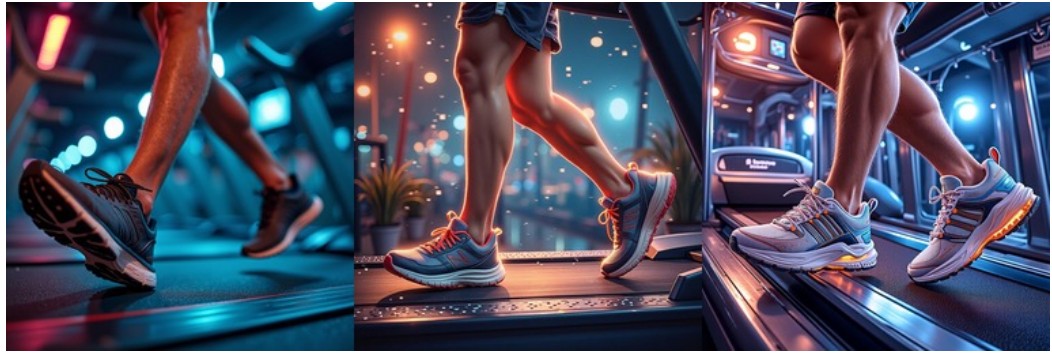

Flux DanceGRPO BranchGRPO(Ours)

Figure 12

Family assembling missile in living room.

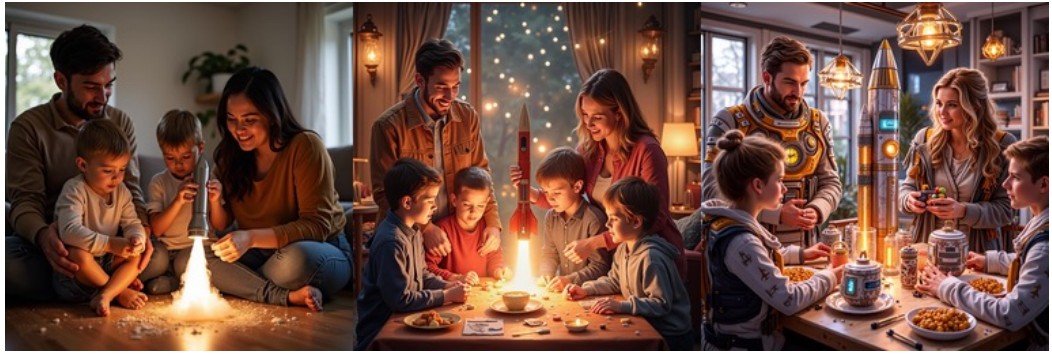

Flux              DanceGRPO              BranchGRPO(Ours)

The image depicts alien flowers and plants surrounded by visceral exoskeletal formations in front of mythical mountains with dramatic contrast lighting, created with surreal hyper detailing in a 3D render.

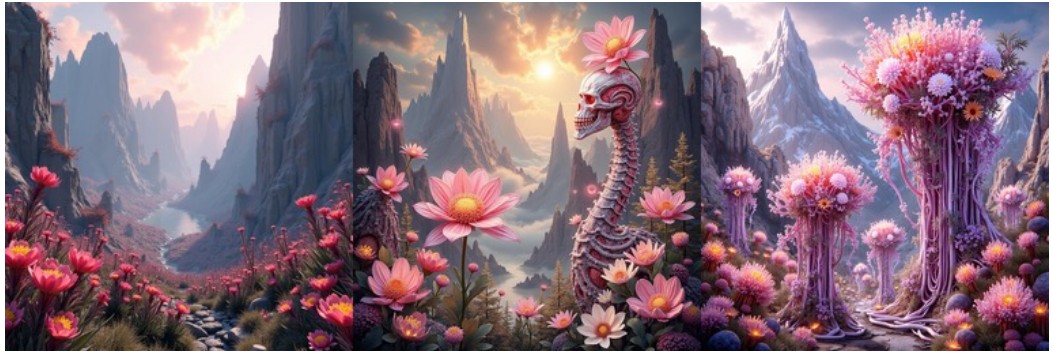

Flux               DanceGRPO              BranchGRPO(Ours)

A colorful tin toy robot runs a steam engine on a path near a beautiful flower meadow in the Swiss Alps with a mountain panorama in the background, captured in a long shot with motion blur and depth of field.

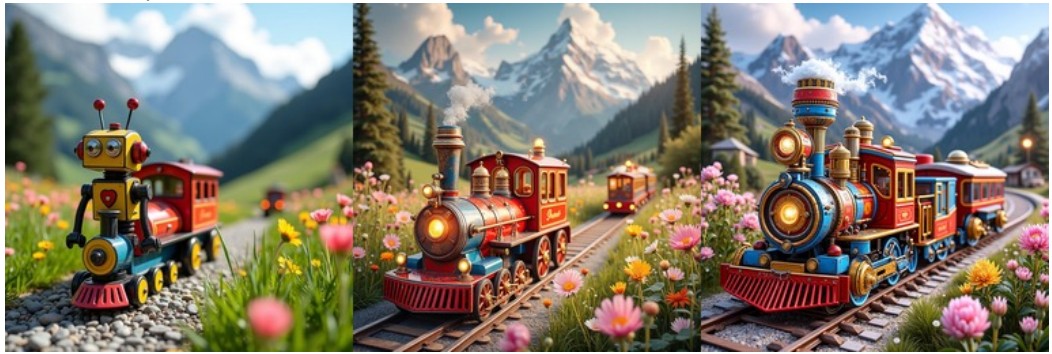

Flux               DanceGRPO              BranchGRPO(Ours)

Figure 13

## C.3   MORE IMAGE2VIDEO RESULTS

Table 13: **Video evaluation on vBench.** We report mean values on 500 samples.

| Method | Aesthetic Quality | Background Consistency | Dynamic Degree | Imaging Quality | Motion Smoothness | Iteration Time (s) |
|---|---|---|---|---|---|---|
| Base Model | 0.5206 | 0.9588 | 0.5150 | 71.92 | 0.9784 | - |
| DanceGRPO | 0.5178 | 0.9647 | 0.4992 | 71.94 | 0.9899 | 1352 |
| BranchGRPO | 0.5190 | 0.9659 | 0.5000 | 71.94 | 0.9912 | 493 |

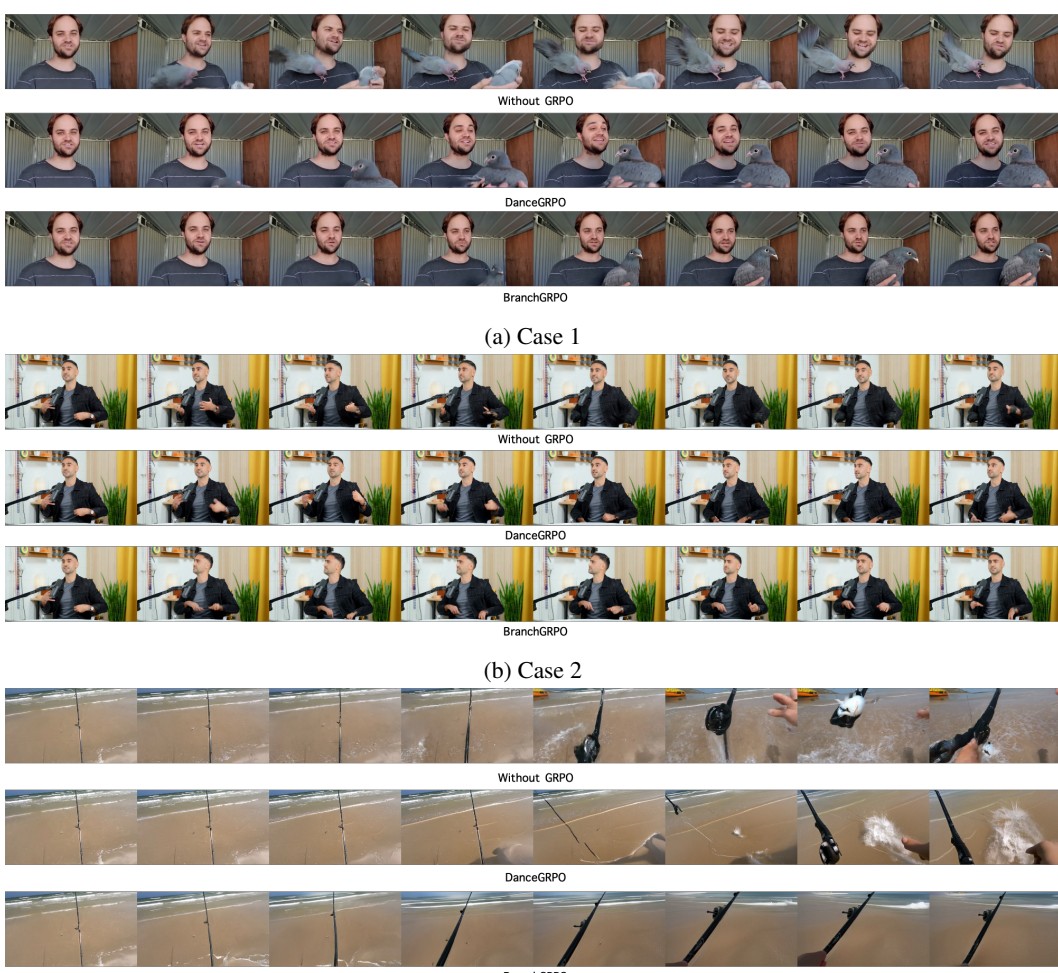

(a) Case 1

(b) Case 2

(c) Case 3

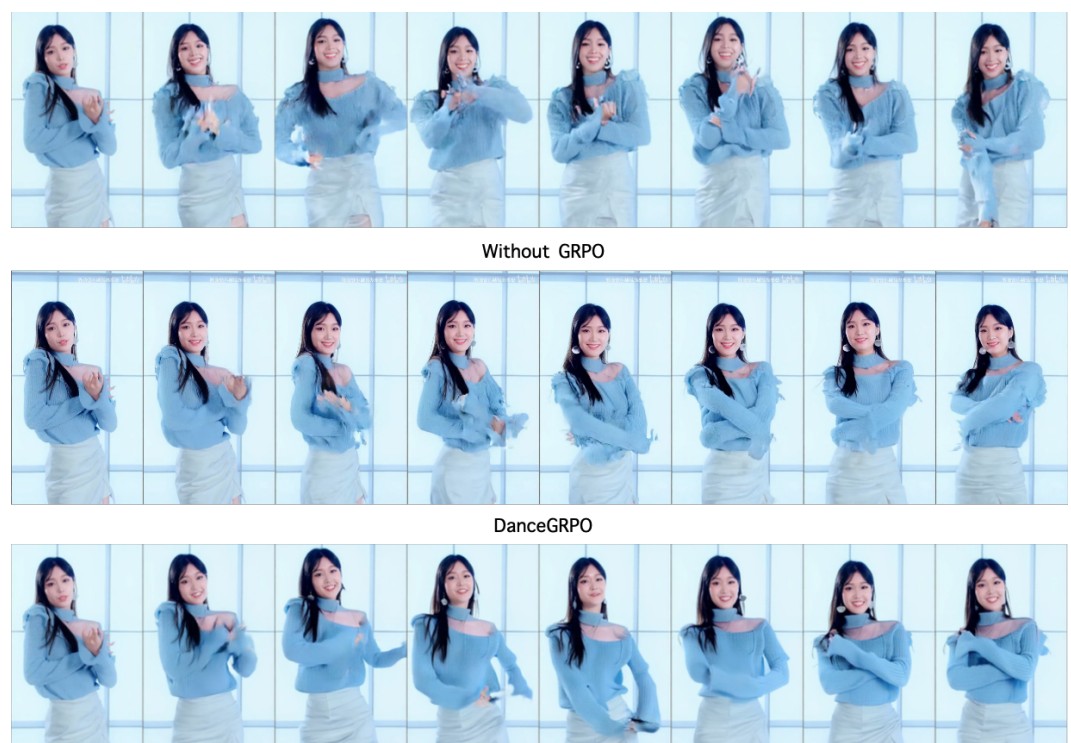

(a) Case 4

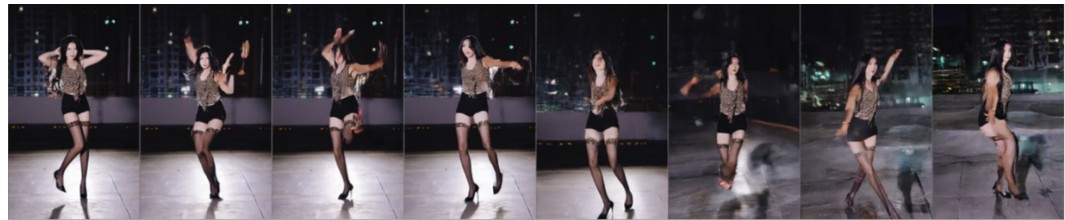

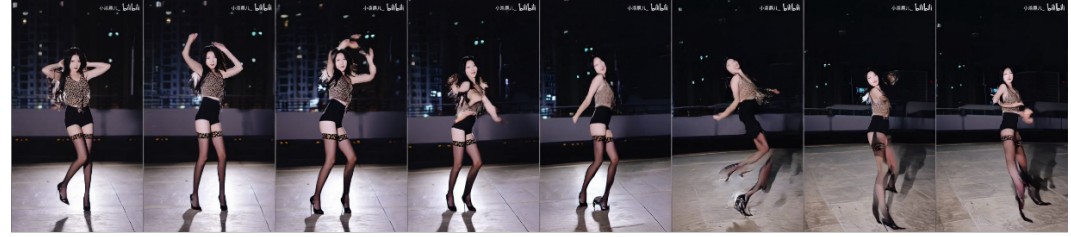

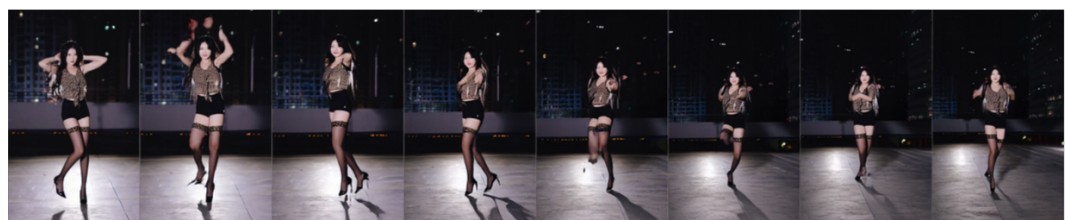

(b) Case 5

## C.4 FAILURE CASES

A colorful digital painting with a front view and anime-inspired vibes featuring a magical composition.

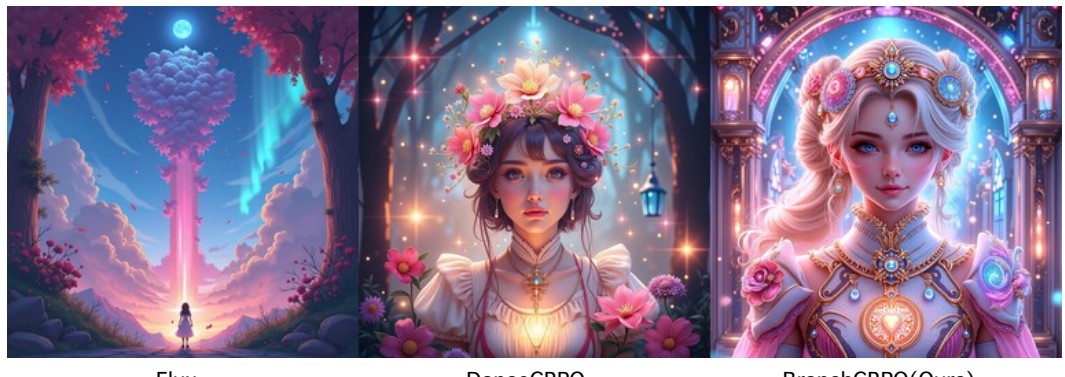

| Flux | DanceGRPO | BranchGRPO(Ours) |

A one-eyed dwarf wizard holding a flagon in clean cel shaded vector art.

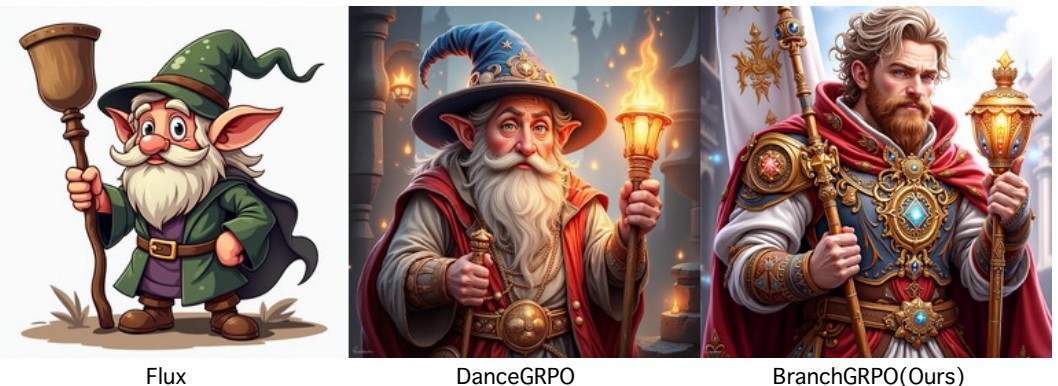

| Flux | DanceGRPO | BranchGRPO(Ours) |

Australian soldiers surrendering to an emu.

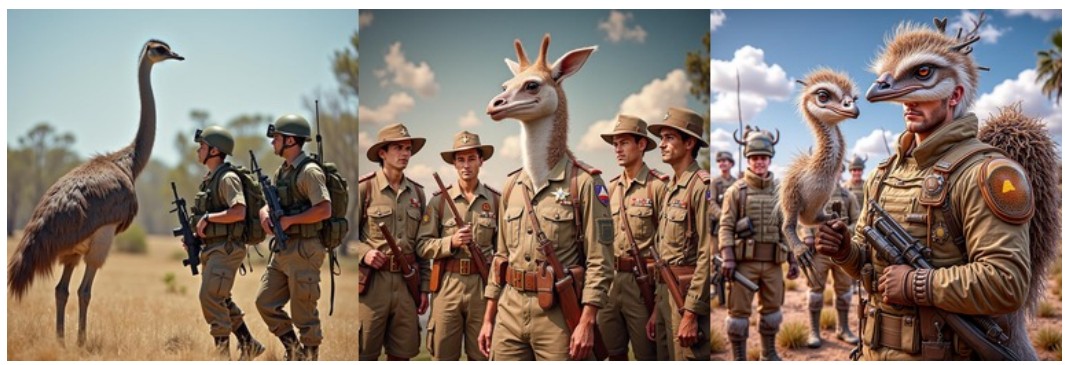

| Flux | DanceGRPO | BranchGRPO(Ours) |

Figure 16: Failure case

Head and shoulders portrait of Jinx from League of Legends of Arcane animated Series.

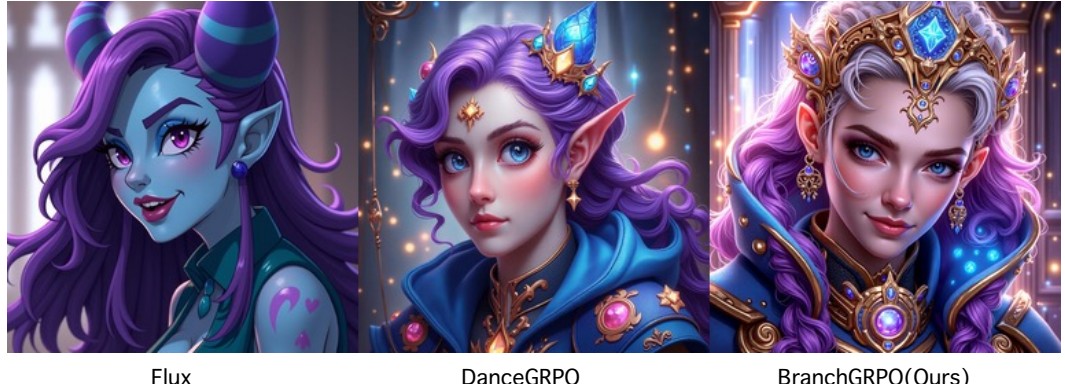

| Flux | DanceGRPO | BranchGRPO(Ours) |

The image is of Pixel Art Huggy Wuggy performing a jumpscare.

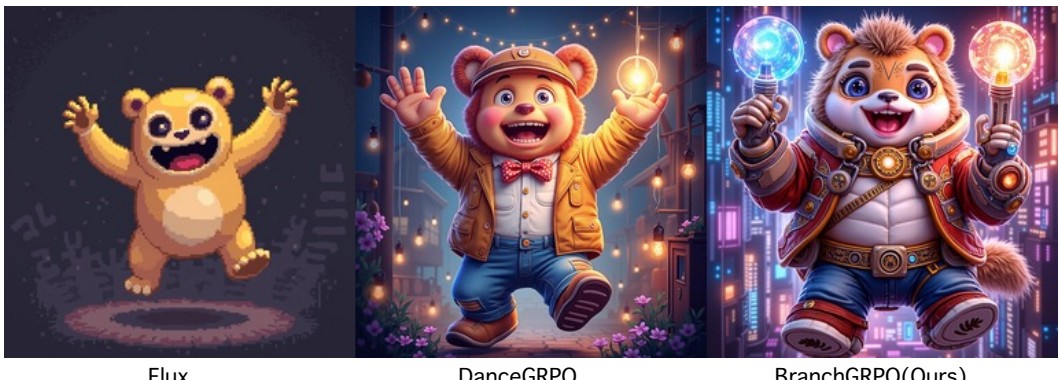

| Flux | DanceGRPO | BranchGRPO(Ours) |

A 3D render of a volcanic icon on a rocky background, in isometric perspective and darkly lit.

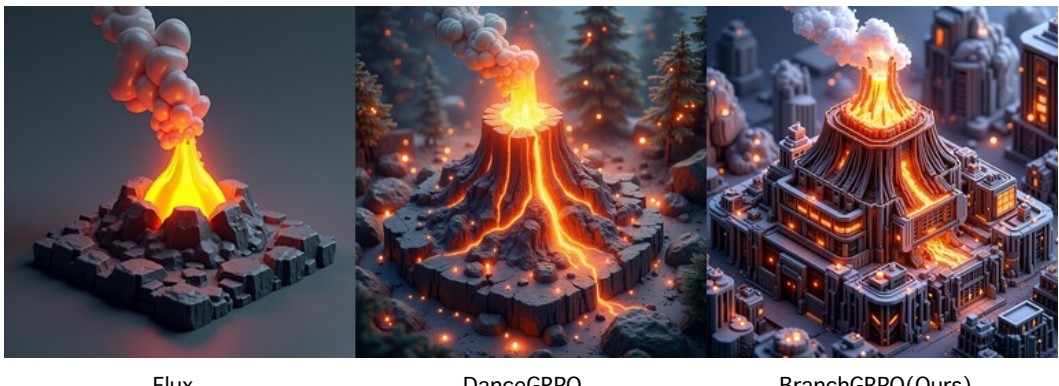

| Flux | DanceGRPO | BranchGRPO(Ours) |

Figure 17: Failure case

# D DISCUSSION AND FUTURE WORK

While our results demonstrate the stability and efficiency benefits of BranchGRPO, several open directions remain.

**Discussion.** BranchGRPO introduces structured branching and pruning into GRPO training, which we have shown to improve both efficiency and alignment. Our ablations suggest that the choice of branching schedule and pruning strategy can substantially affect reward stability, highlighting the importance of principled tree design. Moreover, reward fusion provides stable gradients in practice, but its bias–variance tradeoff under different weighting schemes warrants further theoretical analysis.

**Future Work.** Several promising directions extend beyond the present scope. (1) *Dynamic branching.* Instead of fixed hyperparameters, one can design adaptive policies that adjust branch factor, correlation, or pruning windows on-the-fly based on sample difficulty or intermediate rewards, enabling more efficient rollouts. (2) *Beyond diffusion models.* The branching framework could naturally transfer to other generative paradigms, including diffusion-based LLMs and multimodal foundation models. (3) *Scaling to long-horizon video.* While initial experiments on WanX-1.3B I2V show benefits, more extensive validation on high-resolution, long-duration video generation tasks is required. (4) *Robotics and action generation.* Tree-structured rollouts are naturally suited to robotics, where intermediate states provide dense and verifiable rewards (e.g., task success signals). Extending BranchGRPO to robotic action generation and embodied video generation learning could open a promising direction for embodied AI.

