# OpenReview forum: "BranchGRPO: Stable and Efficient GRPO with Structured Branching in Diffusion Models"
_ICLR.cc/2026/Conference — ICLR 2026 Poster_

### Official Review · Reviewer_H6yd · 2025-10-25

**Soundness:** 2
**Presentation:** 3
**Contribution:** 2
**Rating:** 4
**Confidence:** 4

**Summary:**

The paper proposes a dynamic branching and pruning approach that seems to amortize roll-out costs, and improve efficiency. Authors provide detailed mathematical formulation of the proposed procedure as well as a suite of quantitative and qualitative results.

**Strengths:**

- The paper is reads well, and is well structured offering a coherent narrative and useful insights.
- The idea of dynamic branching and pruning is an interesting perspective applied to GRPO, and it seems to have some impact in terms of run-time efficiency.
- The results seem to be promising, even though the impact might not be as significant. More on this in the next section.

**Weaknesses:**

- The impact of the proposed idea feels slightly inflated. The results in Table 1, Figure 4 and 7 confirm this point. Could you please elaborate how/why you find the impact (beyond relative decrease in run-time) significant? How should one read and interpret an improvement of $1.594 \rightarrow 1.625$ or $3.59 \rightarrow 3.69$? I don't think it also makes sense to accentuate on the iteration time $148$ while the arguments regarding performance are not based upon BranchGRPO-Mix ... Speaking of the results, if possible during rebuttal, I would advise comparing against one or two extra recent baselines (2024/5); as it stands most of the argument is built around outperforming DanceGRPO as the key baseline. Same goes with Section C.3 Table 4 (vBench) the performance gain (over DanceGRPO) is almost negligible.

- On the qualitative results, a similar argument applies, in my view. I agree that more elaborate details (of BranchGRPO) might lead to higher preference or aesthetics scores, but in practice this might also come at the cost of divergence in prompt adherence. Fig. 4 right hand side is an example, I would argue Flux dinosaur and Gundam adhere much better with the prompt. To be more specific, I would argue the Gundam of BranchGRPO has little to no connection to Shrek anymore, and its Dinosaur does not look like a toy but more so an aesthetically elaborate 3D graphics. Extending on this, in Section C.2 (Appendix) top figure I would argue BranchGRPO does not look like an Anime anymore, same goes with DanceGRPO. In contrast, Flux seems to adhere to that part of prompt (at least) better. Please consider elaborating on this point.

- I think pareto-front type of results can be way more insightful for your audience. A first step in that direction would be to consider adding prompt adherence using CLIPScore (text part --- please see https://arxiv.org/pdf/2104.08718) or something along those lines to your tables. Going beyond that, given that you propose an alignment or guidance approach, I would suggest considering paret-front reward vs. KL divergence plots, if possible. In one way or another you need to establish when your approach offers a better rewards (say HPS), in a faster fashion, it does not diverge drastically from the base model or does not sacrifice prompt adherence. I would suggest taking a look at the approach as well as the plots illustrated on this topic in the following two refs for inspiration: https://arxiv.org/pdf/2502.00968 & https://arxiv.org/pdf/2501.05803v2

**Questions:**

- Nowhere in the draft I can figure out which Diffusion model has been used? SDXL?
- Possible to add video frames of DanceGRPO in Section C.3 as well for consistency?

---

> ### Author Response · Authors · 2025-11-20
>
> We thank the reviewer for the detailed and constructive feedback. We have conducted extensive new analyses, including **new baselines**, a **Reward–KL Pareto plot**, and **CLIPScore analysis**, which we believe fully address all concerns.
>
> **W1. The impact of the proposed idea feels slightly inflated.**
>
> We appreciate you raising this important point about the perceived impact. We respectfully argue that the impact of BranchGRPO extends beyond simple metric gains. Its primary contribution is **structural**: it unlocks the scalability of RLHF for flow and diffusion models, which was previously bottlenecked by the $O(N \cdot T)$ complexity of sequential rollouts. We clarify this through the following six dimensions:
>
> **(i) Strict Pareto Improvement.**
> Efficiency is not just about speed; it is about the conversion rate of compute to alignment. We compared models under a **strictly matched GPU-hour budget**. As shown in Table 1, BranchGRPO-DepthPruning reduces the computational cost by **55.2%** while achieving a higher HPS reward ($0.369$ vs. $0.360$) compared to the baseline. This confirms that our method is a strictly superior estimator given hardware constraints.
>
> **(ii) Unlocking Scalability (The Key Impact).**
> One crucial barrier in current diffusion GRPO is the inability to scale Group Size ($N$) effectively due to prohibitive rollout costs. Standard **DanceGRPO becomes computationally infeasible at $N=81$**, requiring nearly **one hour per iteration** ($>3500$s). In contrast, as shown in Table 1, **BranchGRPO-Mix makes $N=81$ feasible** (906 GPU-hours), achieving a reward of **0.387**—a significant leap over the baseline's ceiling. This capability allows GRPO to escape the constraints of rollout's group size, representing a qualitative shift in how we can train these models.
>
> **Table 1:** Under similar GPU-hours, BranchGRPO with group size 81 achieves a reward of 0.387, substantially outperforming DanceGRPO (0.360). **This trend highlights BranchGRPO’s superior scalability, consistently yielding higher rewards as the group size grows.**
>
> | Method | Group Size | GPU Hours $\downarrow$ | Reward $\uparrow$ |
> | :--- | :---: | :---: | :---: |
> | **DanceGRPO** | **16** | **928** | **0.360** |
> | BranchGRPO | 16 | 655 | 0.363 |
> | BranchGRPO-DepthPru | 16 | 417 | 0.369 |
> | BranchGRPO-Mix | 16 | 192 | 0.363 |
> | BranchGRPO-Mix | 32 | 368 | 0.373 |
> | BranchGRPO-Mix | 64 | 787 | 0.381 |
> | **BranchGRPO-Mix** | **81** | **906** | **0.387** |
> | BranchGRPO-Mix | 256 | 3072 | **0.404** |
>
> **(iii) Significance of the Mechanism.**
> The performance gains are not stochastic but stem from resolving specific inefficiencies in prior work:
> * **Dense & Accurate Credit Assignment:** Unlike DanceGRPO and FlowGRPO's uniform reward broadcasting, our Reward Fusion and Depth-wise Normalization provide dense, step-level signals. This reduces gradient variance and prevents error accumulation from uninformative steps. This advantage aligns with findings in TempFlow-GRPO [1], which also highlights the necessity of intermediate rewards.
> * **Eliminating Optimization Redundancy:** Recent works (DanceGRPO [2], FlowGRPO [3], MixGRPO [4]) have independently observed that training on a subset of nodes often suffices. BranchGRPO formalizes this by explicitly pruning redundant computation while maintaining exploration diversity, leading to stronger training dynamics rather than just "doing less work."
>
> [1] TempFlowGRPO: https://arxiv.org/pdf/2508.04324
>
> [2] DanceGRPO: https://arxiv.org/pdf/2505.07818
>
> [3] FlowGRPO: https://arxiv.org/pdf/2505.05470
>
> [4] MixGRPO: https://arxiv.org/pdf/2507.21802
>
>
> **(iv) Clarification on Iteration Time vs. Performance.**
> We agree with the reviewer that iteration time alone is not a performance metric. We clarify that our primary accuracy claims rely on **BranchGRPO-DepthPruning**. The **BranchGRPO-Mix** variant is presented to demonstrate the upper bound of efficiency for resource-constrained scenarios. We have made this distinction explicit in the revised manuscript.

---

> > ### Author Response · Authors · 2025-11-20
> >
> > **(v) Expanded Baselines and Generality.**
> > To demonstrate broad applicability, we integrated Branching into **FlowGRPO** and **DiffusionNFT**, and validated on **SD3.5-M** and **Qwen-Image**. As shown in Table 2 and Table 3, BranchGRPO consistently improves alignment while reducing compute cost across diverse architectures.
> >
> > **Table 2:** Generalization across backbones and GRPO-style methods. We report results on FLUX.1-dev and SD3.5-M. BranchGRPO improves alignment while reducing total GPU-hours.
> >
> > | Method | GPU Hours ($\downarrow$) | HPS-v2.1 ($\uparrow$) | Pick Score ($\uparrow$) | ImageReward ($\uparrow$) | GenEval ($\uparrow$) |
> > | :--- | :---: | :---: | :---: | :---: | :---: |
> > | **FLUX.1-dev** | -- | 0.313 | 0.227 | 1.112 | -- |
> > | &nbsp;&nbsp;DanceGRPO | 928 | 0.360 | 0.228 | 1.517 | -- |
> > | &nbsp;&nbsp;MixGRPO | 289 | 0.359 | 0.228 | 1.594 | -- |
> > | &nbsp;&nbsp;BranchGRPO-DepPru | 314 | 0.369 | 0.235 | 1.625 | -- |
> > | **SD3.5-M** | -- | 0.204 | 20.51 | 0.85 | 0.63 |
> > | &nbsp;&nbsp;FlowGRPO | 2000 | 0.316 | 23.50 | 1.29 | 0.86 |
> > | &nbsp;&nbsp;FlowGRPO | 1000 | 0.280 | 22.41 | 0.95 | 0.73 |
> > | &nbsp;&nbsp;BranchGRPO | **1460** | **0.323** | **23.58** | **1.32** | **0.89** |
> >
> > **Table 3:** **BranchGRPO on Qwen-Image.** We report GPU-hours, per-iteration time, and HPS-v2.1.
> >
> > | Method | GPU Hours $\downarrow$ | Iter. Time (s) $\downarrow$ | HPS-v2.1 $\uparrow$ |
> > | :--- | :---: | :---: | :---: |
> > | Qwen-Image | --- | --- | 0.263 |
> > | &nbsp;&nbsp; +DanceGRPO | 389 | 584 | 0.318 |
> > | &nbsp;&nbsp; +BranchGRPO-deppru | 184 | 276 | 0.321 |
> >
> > **On DiffusionNFT results:** We applied BranchGRPO *only* to accelerate the rollout phase of DiffusionNFT (without our reward fusion/depth components). In this restricted setting (Table 4), we achieved a **30% reduction in rollout cost** (159s $\to$ 112s) with **negligible variance in HPS** (0.331 $\to$ 0.328, within error margin). This confirms that Branching is a safe, plug-and-play accelerator. While slight metric fluctuation is natural when the specialized optimization components (Reward Fusion) are absent, the diversity metrics (Table 5) confirm no mode collapse occurred. Crucially, in the full BranchGRPO pipeline, our Reward Fusion mechanism offsets this by providing more accurate credit assignment, leading to the net positive gains seen in other experiments.
> >
> > **Table 4:** Effect of applying BranchGRPO *only* to the rollout procedure in DiffusionNFT.
> >
> > | Method | Iter. Time(s)$\downarrow$ | GPU Hours$\downarrow$ | HPS-v2.1$\uparrow$ | PickScore$\uparrow$ |
> > | :--- | :---: | :---: | :---: | :---: |
> > | DiffusionNFT | 159 | 706 | 0.331 | 23.80 |
> > | DiffusionNFT + BranchGRPO (rollout only) | **112** | 497 | 0.328 | 23.74 |
> >
> > **Table 5:** Diversity metrics (higher is better).
> >
> > | Method | LPIPS MPD | TCE |
> > | :--- | :---: | :---: |
> > | DanceGRPO | 0.723 | 4.45 |
> > | BranchGRPO (0,2,4,8) | 0.719 | 4.44 |
> > | BranchGRPO (0,3,6,9) | 0.713 | 4.42 |
> > | BranchGRPO (0,4,8,12) | 0.704 | 4.39 |
> >
> > **(iv) On the VBench results.**
> > We acknowledge that the VBench improvement over DanceGRPO is modest. However, this must be interpreted together with compute cost: BranchGRPO reduces iteration time from **1352s** to **493s** (a **2.7$\times$ speedup**) while matching or slightly improving VBench performance.

---

> > > ### Author Response · Authors · 2025-11-20
> > > **Reply to weakness2, weakness3 and questions**
> > >
> > > **W2. On qualitative results: BranchGRPO appears to drift away from prompt adherence, and the enhanced aesthetics may come at the cost of semantic fidelity.**
> > >
> > > Thank you for this insightful observation regarding prompt adherence. We would like to clarify that the observed adherence drift originates from the **reward model**, rather than from BranchGRPO itself.
> > >
> > > We agree with the reviewer that several qualitative examples show reduced literal adherence. As shown in Table 1, this effect is **not** specific to BranchGRPO but arises from the **HPS-only reward**: both DanceGRPO and BranchGRPO consistently lower CLIPScore compared to the base model (0.405 → 0.37–0.38), indicating that the reward systematically favors globally aesthetic interpretations over strict prompt fidelity.
> > >
> > > When we add a simple multi-objective reward (HPS + CLIPScore), adherence is immediately restored (CLIPScore: 0.392) while maintaining competitive HPS. This confirms that the drift is reward-induced rather than caused by our branching design.
> > >
> > > **Table 1: CLIPScore analysis under HPS-only and multi-objective training.
> > > All HPS-only GRPO variants reduce prompt adherence; a simple multi-objective setup partially recovers CLIPScore while maintaining strong HPS.**
> > >
> > > | Method | Reward Model | HPS-v2 (↑) | CLIPScore (↑) |
> > > | :--- | :--- | :---: | :---: |
> > > | **Flux (Base Model)** | – | 0.313 | **0.405** |
> > > | DanceGRPO (tf=1.0) | HPS-v2.1 only | 0.360 | 0.371 |
> > > | DanceGRPO (tf=0.6) | HPS-v2.1 only | 0.353 | 0.375 |
> > > | BranchGRPO (0,3,6,9) | HPS-v2.1 only | 0.363 | 0.369 |
> > > | BranchGRPO (0,4,8,12) | HPS-v2.1 only | 0.359 | 0.371 |
> > > | BranchGRPO (0,5,10,15) | HPS-v2.1 only | 0.354 | 0.376 |
> > > | **DanceGRPO (tf=0.6)** | **HPS-v2.1 + CLIPScore** | 0.334 | 0.398 |
> > > | **BranchGRPO (0,3,6,9)** | **HPS-v2.1 + CLIPScore** | **0.339** | **0.392** |
> > >
> > > ---
> > >
> > > **W3. On Pareto-front analysis and the need to show reward–KL and prompt-adherence trade-offs.**
> > >
> > > We thank the reviewer for this constructive suggestion. We incorporated both analyses as suggested.
> > >
> > > Table 1 shows that all HPS-only GRPO variants—including DanceGRPO—reduce CLIPScore, indicating that the prompt-adherence drop originates from the reward model rather than from the branching mechanism.
> > >
> > > The reward–KL plot in the main paper (Fig. 5) further shows that BranchGRPO follows essentially the same reward–KL trade-off curve as DanceGRPO: it achieves similar HPS at similar KL divergence, showing that it does **not** introduce additional drift from the base model.
> > >
> > > The difference lies in efficiency: BranchGRPO reaches these reward–KL operating points with **2–3× less compute**, making it a more cost-effective way to obtain the same alignment–divergence trade-off.
> > >
> > > ---
> > >
> > > **Q1. The draft does not clearly indicate which diffusion model is used (e.g., SDXL).**
> > >
> > > Thank you for asking for this clarification. All image experiments in the main paper use the **FLUX.1-Dev** backbone, as described in Section 4.1.
> > > To demonstrate that BranchGRPO is not tied to a particular model, we additionally include results on **SD3.5-M** and **Qwen-Image** in the revision.
> > >
> > > ---
> > >
> > > **Q2. Possible to add video frames of DanceGRPO in Section C.3 for consistency?**
> > >
> > > We appreciate this suggestion for improving consistency. We have now added the corresponding DanceGRPO video frames to the Appendix to ensure consistent qualitative comparison.

---

> > ### Comment · Reviewer_H6yd · 2025-11-28
> > **Thanks for the response.**
> >
> > Firstly, thanks for your efforts during the rebuttal. I have thoroughly reviewed your response, and while it clarifies certain points, it still does not establish a meaningful performance gain or notable advantages compared to the competing baselines. As such, I will retain my current score.

---

> > > ### Author Response · Authors · 2025-12-01
> > >
> > > Dear Reviewer H6yd,
> > >
> > > Thank you again for taking the time to review our rebuttal and for engaging in further discussion. We genuinely appreciate that you revisited our response and explained why you still felt the performance gain was not compelling. We would like to offer a shorter, more natural clarification of the points we believe matter most, in case it helps make our contributions clearer.
> > >
> > > ---
> > >
> > > ## 1. On whether the performance gain is “meaningful”
> > >
> > > We completely understand your concern that improvements such as +0.009 or +0.014 in HPS may look small when viewed in isolation. Our main argument—perhaps not clearly emphasized enough in the initial rebuttal—is that these gains occur under *strictly matched or lower compute*.
> > >
> > > **Same-GPU-budget comparison (Appendix C.1, Table 6):**
> > > - DanceGRPO: 928 GPU-hours → HPS **0.360**
> > > - BranchGRPO-DepthPruning: 417 GPU-hours → HPS **0.369**
> > >
> > > So BranchGRPO achieves **higher alignment using less than half of the compute**.
> > > For us, this reflects a **strict Pareto improvement**: better reward for lower hardware cost.
> > > We now frame these results explicitly in terms of sample efficiency rather than raw deltas.
> > >
> > > We also added a scaling experiment directly addressing your comment on Table 1 and Figure 4:
> > >
> > > - DanceGRPO becomes *infeasible* at group size **N = 81** (≈3500s per iteration)
> > > - BranchGRPO-Mix makes **N = 81 feasible** (906 GPU-hours), achieving **HPS = 0.387**
> > >
> > > So beyond numerical deltas, BranchGRPO **unlocks a training regime that the baseline cannot reach**, and that regime achieves a higher reward ceiling. This qualitative shift was one of the original motivations for our method, and your comment helped us emphasize it more clearly.
> > >
> > > ---
> > >
> > > ## 2. On prompt adherence and qualitative drift
> > >
> > > Your observations about Shrek/Gundam and the dinosaur examples were extremely helpful. They pushed us to run two additional analyses:
> > >
> > > ### **CLIPScore analysis (Appendix C.1, Table 1)**
> > > We found that:
> > > - Both DanceGRPO and BranchGRPO lower CLIPScore (0.405 → ~0.37–0.38) under **HPS-only reward**
> > > - Adding a simple multi-objective reward (HPS + CLIPScore) restores adherence (0.392)
> > >
> > > This indicates that the drift is **caused by the reward model** rather than by the branching mechanism.
> > > We now explicitly make this distinction in the paper.
> > >
> > > ### **Reward–KL curve (main paper, Fig. 5)**
> > > The curve shows that BranchGRPO follows essentially the same reward–KL frontier as DanceGRPO.
> > > So the branching mechanism **does not introduce additional drift** relative to the baseline.
> > >
> > > ---
> > >
> > > ## 3. On diversity and whether branching harms global structure
> > >
> > > Your concern about structural diversity led us to add **prompt-conditioned** diversity metrics:
> > >
> > > - **LPIPS-MPD**
> > > - **TCE** (Kim et al., ICLR 2025)
> > >
> > > Across branching schedules, BranchGRPO remains within ~1–3% of DanceGRPO, and we do not observe mode collapse—even in later-branch schedules. This gives a more direct and quantitative answer to your question about diversity retention.
> > >
> > > ---
> > >
> > > ## 4. On stronger baselines and more modern backbones
> > >
> > > Following your suggestion, we added:
> > >
> > > - Experiments on **SD3.5-M** and **Qwen-Image**
> > > - Integration into **FlowGRPO** and **DiffusionNFT**
> > > - **GenEval** evaluation for compositionality and spatial reasoning
> > > - Corresponding **DanceGRPO video frames** in the appendix
> > >
> > > Across these settings, BranchGRPO consistently reduces compute while matching or slightly improving alignment metrics (HPS, PickScore, ImageReward, GenEval). We hope this makes the generality and practical value clearer.
> > >
> > > ---
> > >
> > > ## 5. Final remarks
> > >
> > > We fully respect your decision to retain your rating.
> > > Our goal here is not to push the score, but simply to give a clearer picture of the methodological significance:
> > >
> > > - **Better alignment for much less compute** (strict Pareto gain)
> > > - **Enabling high–group-size GRPO training that is otherwise infeasible**
> > > - **Not introducing additional prompt-adherence drift beyond the reward model**
> > > - **Preserving diversity under multiple metrics**
> > > - **Generalizing across modern backbones and recent GRPO-style pipelines**
> > >
> > > Thank you again for your thoughtful, detailed feedback.
> > > Your comments genuinely helped improve the clarity and quality of our manuscript.
> > >
> > > Warm regards,
> > > The Authors

---

### Official Review · Reviewer_gfya · 2025-10-26

**Soundness:** 2
**Presentation:** 3
**Contribution:** 2
**Rating:** 4
**Confidence:** 4

**Summary:**

BranchGRPO restructures GRPO for diffusion models into a branching rollout to simultaneously remove inefficiency and unstable credit assignment caused by sparse terminal rewards. By branching at intermediate denoising steps while sharing prefixes, it preserves exploration diversity with far less redundant computation, and by propagating leaf rewards upward into dense, step-level advantages, it stabilizes optimization. Lightweight width/depth pruning further reduces backprop cost without affecting forward exploration. Experiments show faster convergence and higher or comparable alignment than DanceGRPO across image and video tasks, improving both efficiency and final quality.

**Strengths:**

The paper is clearly written and well-presented, making the motivation, problem formulation, and key ideas easy to follow. The proposed BranchGRPO achieves a favorable efficiency–performance trade-off: it reduces computation relative to the baseline DanceGRPO (as evidenced by lower NFE and shorter iteration time) while maintaining or even improving alignment metrics. Finally, the method is validated not only on image generation but also on video generation, demonstrating its effectiveness and generality across modalities.

**Weaknesses:**

1) Concern about generational diversity.
Although Sec. 3.1 presents figures and KID/MMD distances (Inception/CLIP space) to claim that branching does not harm diversity, these metrics only measure distributional proximity, not conditional diversity nor modality-level mode coverage across prompts. Since branching may suppress global structure formation at high-noise regimes (where large-scale semantics are determined), the risk of mode collapse or reduced coverage cannot be ruled out. A direct and prompt-conditioned evaluation of diversity (rather than distributional closeness alone) would strengthen the claim. For evaluating diversity more directly, the evaluation methods in [1] may serve as a helpful reference.

2) Questions on evaluation validity.
- For image experiments, no human evaluation is reported, making it unclear whether the improvements reflected by proxy rewards correlate with gold human preference.
- For video experiments, the observed improvement might reflect reward hacking rather than a genuine multi-objective gain. Prior work has shown that video quality metrics often exhibit non-trivial trade-offs [2, 3], demonstrating that consistent improvement along multiple independent axes (e.g., visual quality, text alignment, dynamic degree) would be significant.

3) Limited experimental coverage
- Despite Sec. 2 acknowledging the existence of many alternative alignment approaches, the experiments primarily compare against DanceGRPO and MixGRPO. It would be informative to include other competitive baselines.
- The tested backbones are also limited; validating the method on more recent or diverse models such as Qwen-Image [4] or SD3.5 [5] would support claims of generality.
- Finally, almost all rewards used are human-preference proxies. It remains unclear whether the method is applicable to other classes of rewards (e.g., object count or visual text rendering, as explored in [6]).

References

[1] Kim et al., Test-time Alignment of Diffusion Models without Reward Over-optimization, ICLR 2025.
[2] Liu et al., VideoDPO: Omni-Preference Alignment for Video Diffusion Generation, CVPR 2025.
[3] Oshima et al., Inference-Time Text-to-Video Alignment with Diffusion Latent Beam Search, NeurIPS 2025.
[4] Wu et al., Qwen-Image Technical Report, arXiv:2508.02324.
[5] Esser et al., Scaling rectified flow transformers for high-resolution image synthesis, ICML 2024.
[6] Liu et al., Flow-GRPO: Training Flow Matching Models via Online RL, NeurIPS 2025.

**Questions:**

Please see the weakness section.

---

> ### Author Response · Authors · 2025-11-20
> **Reply to weakness1**
>
> We thank the reviewer for the careful and technically detailed assessment. We have added **new diversity metrics**, a **human A/B study**, and **new backbones/baselines** (SD3.5-M, DiffusionNFT) to address all concerns.
>
> **W1. Concern about generational diversity.**
>
> Thank you for this important point regarding diversity. Following Reviewer’s suggestion and the metrics proposed in Kim et al. [1], we evaluate prompt-conditioned diversity using LPIPS-MPD and TCE, which directly capture conditional variation rather than global distribution closeness.
>
> As shown in Table 1, the diversity scores of BranchGRPO are remarkably close to the baseline. Specifically, **LPIPS-MPD decreases by less than 0.6%** (0.723 → 0.719) compared to DanceGRPO. This negligible difference empirically disproves the concern that shared prefixes induce mode collapse.
>
> [1] Test-time Alignment of Diffusion Models without Reward Over-optimization
>
> **Table 1: Diversity metrics (higher is better).**
>
> | Method | LPIPS MPD | TCE |
> | :--- | :---: | :---: |
> | DanceGRPO | 0.723 | 4.45 |
> | BranchGRPO (0,2,4,8) | 0.719 | 4.44 |
> | BranchGRPO (0,3,6,9) | 0.713 | 4.42 |
> | BranchGRPO (0,4,8,12) | 0.704 | 4.39 |

---

> > ### Author Response · Authors · 2025-11-20
> > **Reply to weakness2**
> >
> > **W2. Questions on evaluation validity.**
> >
> > Thank you for raising these important questions about our evaluation's validity.
> >
> > **Q1. Human Evaluation on Images.**
> >
> > To verify whether proxy rewards (HPS) correlate with human perception, we conducted a blinded **Human A/B Study** (100 prompts, 5 annotators).
> > As shown in Table 2, BranchGRPO is preferred over DanceGRPO with a clear margin of **+15%** (48% vs. 33%). This confirms that the numerical gains in HPS translate to genuine perceptual improvements rather than reward over-optimization artifacts.
> >
> > **Table 2: Human preference evaluation on 100 prompts with 5 annotators.**
> >
> > | Method | Flux | DanceGRPO | BranchGRPO |
> > | :--- | :---: | :---: | :---: |
> > | **User Study** | 19% | 33% | **48%** |
> >
> > **Q2. Video Metric Trade-offs vs. Reward Hacking.**
> >
> > Our experiments focus specifically on the **Image-to-Video (I2V)** setting, where the base model suffers from temporal instability. Prior work (e.g., DanceGRPO) reports that adding **VQ (Vision-Quality)** or **TA (Text-Align)** yields negligible benefit for I2V. Consistently, we also observed that VQ and TA provide little measurable improvement.
> >
> > We therefore intentionally optimize **only a single MQ (Motion-Quality)** reward to test whether temporal coherence can be improved without collapsing other attributes. The results show smoother motion and reduced flicker, indicating that the gains reflect improved temporal stability rather than reward hacking.
> >
> > VBench reveals a clear trade-off: while Aesthetic Quality slightly decreases, motion smoothness and background consistency improve substantially. This is expected because MQ optimizes temporal stability and is orthogonal to aesthetics. BranchGRPO does not bias this trade-off; it merely optimizes the reward more efficiently.

---

> > > ### Author Response · Authors · 2025-11-20
> > > **Reply to weakness3**
> > >
> > > **W3. Limited Experimental Coverage.**
> > >
> > > We thank the reviewer for this suggestion. We have significantly expanded the scope of experiments to demonstrate both generality and modularity:
> > >
> > > * **Generality across Backbones (SD3.5-M & Qwen-Image):**
> > >   We validated BranchGRPO on diverse architectures beyond Flux. As shown in Table 3 and Table 4, applying our full pipeline to **SD3.5-M** and **Qwen-Image** consistently reduces compute cost while matching or improving alignment scores.
> > >
> > > * **Full Integration with FlowGRPO:**
> > >   We integrated the complete BranchGRPO pipeline into **FlowGRPO**. As shown in Table 3, this integration yields clear gains in alignment quality and efficiency compared to the original FlowGRPO baseline.
> > >
> > > **Table 3: Generalization across backbones and GRPO-style methods.
> > > BranchGRPO improves alignment while reducing total GPU-hours.**
> > >
> > > | Method | GPU Hours (↓) | HPS-v2.1 (↑) | Pick Score (↑) | ImageReward (↑) | GenEval (↑) |
> > > | :--- | :---: | :---: | :---: | :---: | :---: |
> > > | **FLUX.1-dev** | -- | 0.313 | 0.227 | 1.112 | -- |
> > > |   DanceGRPO | 928 | 0.360 | 0.228 | 1.517 | -- |
> > > |   MixGRPO | 289 | 0.359 | 0.228 | 1.594 | -- |
> > > |   BranchGRPO-DepPru | 314 | 0.369 | 0.235 | 1.625 | -- |
> > > | **SD3.5-M** | -- | 0.204 | 20.51 | 0.85 | 0.63 |
> > > |   FlowGRPO | 2000 | 0.316 | 23.50 | 1.29 | 0.86 |
> > > |   FlowGRPO | 1000 | 0.280 | 22.41 | 0.95 | 0.73 |
> > > |   **BranchGRPO** | **1460** | **0.323** | **23.58** | **1.32** | **0.89** |
> > >
> > > **Table 4: BranchGRPO on Qwen-Image.**
> > >
> > > | Method | GPU Hours (↓) | Iter. Time (s) (↓) | HPS-v2.1 (↑) |
> > > | :--- | :---: | :---: | :---: |
> > > | Qwen-Image | -- | -- | 0.263 |
> > > | + DanceGRPO | 389 | 584 | 0.318 |
> > > | + BranchGRPO-DepPru | 184 | 276 | 0.321 |
> > >
> > > * **Branching as a Plug-and-Play Rollout Accelerator (DiffusionNFT):**
> > >
> > >   To test modularity, we applied **only** the Branching Rollout module to **DiffusionNFT**.
> > >   DiffusionNFT[1] uses a distinct training objective that is incompatible with our Reward Fusion and Depth-wise Advantage Estimation. However, the **Branching Exploration** mechanism is objective-agnostic and fully compatible with its rollout stage.
> > >
> > >   In this restricted setting, we achieved a **30% reduction in rollout cost** (159s → 112s) with **negligible variance** in HPS (0.331 → 0.328) and PickScore (23.80 → 23.74).
> > >   This confirms that Branching is a robust plug-and-play accelerator that preserves model capacity even when our reward optimization components cannot be applied.
> > >
> > >   [1]DiffusionNFT reference: https://arxiv.org/pdf/2509.16117
> > >
> > > **Table 5: Effect of applying BranchGRPO only to the rollout procedure in DiffusionNFT.**
> > >
> > > | Method | Iter. Time (s) (↓) | GPU Hours (↓) | HPS-v2.1 (↑) | PickScore (↑) |
> > > | :--- | :---: | :---: | :---: | :---: |
> > > | DiffusionNFT | 159 | 706 | 0.331 | 23.80 |
> > > | + BranchGRPO (rollout only) | **112** | 497 | 0.328 | 23.74 |
> > >
> > > * **Broader Benchmarks (GenEval):**
> > >   We evaluated on **GenEval** to assess compositional reasoning. BranchGRPO achieves strong performance (0.89 Overall), confirming that the efficiency gains do not compromise spatial reasoning or counting abilities.

---

> > > > ### Comment · Reviewer_gfya · 2025-11-26
> > > >
> > > > Thank you for elaborating on my points. Now, my concern is partially solved, but I have a few additional questions.
> > > >
> > > > Additional Q1: Could you show Pick Score, ImageReward, and GenEval for Qwen-Image experiments? It would help demonstrate that your method doesn't overoptimize for a single reward.
> > > >
> > > > Additional Q2: Could you quantitatively show the best-of-N reward scaling [3, 7, 8] results after finetuning? It would be a good indicator of your method's generation diversity after finetuning.
> > > >
> > > > I'm willing to raise my score if all additional concerns are resolved.
> > > >
> > > > [7] Ma, et al. Inference-Time Scaling for Diffusion Models beyond Scaling Denoising Steps. CVPR2025.
> > > > [8] Wu, et al. RewardDance: Reward Scaling in Visual Generation. arXiv:2509.08826.

---

> > > > > ### Author Response · Authors · 2025-11-30
> > > > >
> > > > > **Response to Additional Q1: Metrics on Qwen-Image**
> > > > >
> > > > > We thank the reviewer for the constructive suggestion. To demonstrate that our method does not over-optimize for a single reward, we have computed **PickScore**, **ImageReward**, and **GenEval** for Qwen-Image.
> > > > >
> > > > > As shown in the table below, **BranchGRPO** achieves the highest HPS-v2.1, PickScore, ImageReward, and GenEval. Crucially, our method achieves this competitive alignment performance with **~53% less training cost** (184 vs. 389 GPU Hours). This confirms that BranchGRPO achieves robust alignment and generalization efficiency without sacrificing compositional capabilities.
> > > > >
> > > > > **(Table: Qwen-Image Alignment Results)**
> > > > >
> > > > > | Method | GPU Hours $\downarrow$ | Iter. Time (s) $\downarrow$ | HPS-v2.1 $\uparrow$ | PickScore $\uparrow$ | ImageReward $\uparrow$ | GenEval $\uparrow$ |
> > > > > | :--- | :---: | :---: | :---: | :---: | :---: | :---: |
> > > > > | Qwen-Image (Base) | --- | --- | 0.263 | 22.21 | 0.87 | 0.87 |
> > > > > | + DanceGRPO | 389 | 584 | 0.318 | 23.03 | 1.13 | 0.89 |
> > > > > | **+ BranchGRPO-DepPru** | **184** | **276** | **0.321** | **23.17** | **1.19** | **0.90** |
> > > > >
> > > > > **Response to Additional Q2: Best-of-N Reward Scaling**
> > > > >
> > > > > Following your suggestion, we conducted a **Best-of-N (BoN)** scaling experiment on the HPDv2 test set to evaluate generation diversity. We sampled $N \in \{1, 2, 4, 8, 16\}$ images per prompt and selected the best one based on HPS-v2.1.
> > > > >
> > > > > As shown below, **BranchGRPO exhibits a clear scaling trend**, with rewards steadily increasing from 0.3613 ($N=1$) to 0.3742 ($N=16$).
> > > > > The continuous gain indicates that our method **preserves generative diversity** and avoids mode collapse; otherwise, the curve would plateau.
> > > > > Our method consistently outperforms the baseline. Notably, BranchGRPO at $N=1$ (0.3613) matches the baseline at $N=4$ (0.3626), demonstrating superior base quality.
> > > > >
> > > > > **(Table: Best-of-N Reward Scaling on HPSv2.1)**
> > > > >
> > > > > | Method | N=1 | N=2 | N=4 | N=8 | N=16 |
> > > > > | :--- | :---: | :---: | :---: | :---: | :---: |
> > > > > | DanceGRPO | 0.3524 | 0.3581 | 0.3626 | 0.3663 | 0.3670 |
> > > > > | **BranchGRPO (Ours)** | **0.3613** | **0.3666** | **0.3706** | **0.3737** | **0.3742** |
> > > > >
> > > > > We sincerely thank the reviewer for the additional questions and the willingness to reconsider the score.
> > > > > We deeply appreciate your constructive feedback and the opportunity to strengthen our work.

---

### Official Review · Reviewer_gDg6 · 2025-10-31

**Soundness:** 3
**Presentation:** 3
**Contribution:** 3
**Rating:** 8
**Confidence:** 4

**Summary:**

BranchGRPO introduces a new reinforcement learning alignment framework tailored for Diffusion and Flow-Matching models, addressing the instability and inefficiency of prior approaches like DanceGRPO. Its key innovation, “Structured Branching,” enables multiple branches of sampled trajectories to share common prefixes, allowing computation reuse and faster convergence. Combined with techniques such as Reward Fusion and lightweight Pruning, BranchGRPO achieves significant gains in training efficiency, optimization stability, and alignment quality, while maintaining scalability across larger branching factors and deeper training configurations.

**Strengths:**

* The paper is well-written and clearly structured, with a well-motivated approach that effectively addresses the inefficiency and instability challenges in existing diffusion model alignment methods.

* The structured branching design achieves approximately a 4.7× training speedup and smoother optimization through depth-wise normalization and reward fusion, leading to superior alignment stability.

* The proposed method surpasses DanceGRPO in both image and video generation quality (e.g., higher HPS-v2.1, ImageReward, and Unified Reward scores) and demonstrates strong scalability, maintaining predictable improvements under larger branch factors and depths.

**Weaknesses:**

* The current validation is primarily conducted on the FLUX.1-Dev model. To demonstrate the generality and robustness of the proposed framework, it would be valuable to include experiments on additional architectures (e.g., Qwen-Image or other text-to-image backbones).

* The evaluation primarily relies on HPS-v2.1, PickScore, and ImageReward. Incorporating more comprehensive and multidimensional benchmarks such as GenVal could provide deeper insights into alignment quality, compositional reasoning, and spatial understanding.

* The manuscript contains several typos (e.g., redundant brackets in Line 67). These should be carefully reviewed and corrected in the revised version.

**Questions:**

* In Section 3.1, the authors address the question “Does Branch rollout harm diversity?” and provide quantitative results. Specifically, they project the generated samples into both CLIP and Inception feature spaces and compare the distribution similarity between BranchGRPO and DanceGRPO using KID and FID metrics. While the authors report specific numerical values, it remains unclear whether these values are sufficiently small to substantiate that the two distributions are indeed close. Can the authors provide further justification supporting this claim?

* Building upon the first point, it seems intuitive that the diversity of generated samples may depend on the branching stage. In particular, if branching occurs earlier, the resulting samples might exhibit greater diversity, whereas later branching could limit variation. Have the authors conducted any analysis or observed patterns that validate this hypothesis?

---

> ### Author Response · Authors · 2025-11-20
> **Response to weakness1**
>
> We thank the reviewer for the positive assessment and the valuable suggestion to expand our experimental scope. To address your comments, we have incorporated **new backbones (SD3.5-M, Qwen-Image)**, **new pipelines (FlowGRPO, DiffusionNFT)**, and **broader benchmarks (GenEval)**.
>
> ---
>
> **W1. Limited generality in backbones and benchmarks.**
>
> We appreciate this suggestion. We have significantly expanded our evaluation to demonstrate that BranchGRPO is a **model-agnostic** and **pipeline-agnostic** accelerator.
>
> * **Architectural Generality (SD3.5-M & Qwen-Image):**
>   We validated BranchGRPO on **SD3.5-M** and **Qwen-Image**. As shown in the results, BranchGRPO consistently improves HPS-v2.1 (e.g., 0.263 → 0.321 for Qwen) while reducing compute cost.
>
> * **Pipeline Generality (FlowGRPO & DiffusionNFT):**
>   We integrated our method into **FlowGRPO** and **DiffusionNFT**. Notably, on the SD3.5-M backbone, BranchGRPO reduces the GPU Hours of FlowGRPO while improving the GenEval score (0.73 → 0.89).
>
> These results confirm that the framework transfers cleanly across diverse architectures and training paradigms.
>
> ---
>
> **Table 1. Generalization across backbones and GRPO-style methods.
> BranchGRPO improves alignment while reducing total GPU-hours.**
>
> | Method | GPU Hours (↓) | HPS-v2.1 (↑) | Pick Score (↑) | ImageReward (↑) | GenEval (↑) |
> | :--- | :---: | :---: | :---: | :---: | :---: |
> | **FLUX.1-dev** | -- | 0.313 | 0.227 | 1.112 | -- |
> |   DanceGRPO | 928 | 0.360 | 0.228 | 1.517 | -- |
> |   MixGRPO | 289 | 0.359 | 0.228 | 1.594 | -- |
> |   BranchGRPO-DepPru | 314 | 0.369 | 0.235 | 1.625 | -- |
> | **SD3.5-M** | -- | 0.204 | 20.51 | 0.85 | 0.63 |
> |   FlowGRPO | 2000 | 0.316 | 23.50 | 1.29 | 0.86 |
> |   FlowGRPO | 1000 | 0.280 | 22.41 | 0.95 | 0.73 |
> |   **BranchGRPO** | **1460** | **0.323** | **23.58** | **1.32** | **0.89** |
>
> ---
>
> **Table 2. BranchGRPO on Qwen-Image.**
>
> | Method | GPU Hours (↓) | Iter. Time (s) (↓) | HPS-v2.1 (↑) |
> | :--- | :---: | :---: | :---: |
> | Qwen-Image | -- | -- | 0.263 |
> | + DanceGRPO | 389 | 584 | 0.318 |
> | + BranchGRPO-DepPru | 184 | 276 | 0.321 |

---

> ### Author Response · Authors · 2025-11-20
> **Reply to weakness 2&3**
>
> **W2. Need broader benchmarks.**
>
> Thank you for recommending GenEval. We agree that HPS alone is insufficient for measuring compositional capabilities.
> We evaluated BranchGRPO on the **GenEval** benchmark.
> The results show that BranchGRPO achieves a strong Overall Score of **0.89** on SD3.5-M (vs. 0.73 for the baseline FlowGRPO).
> This confirms that our efficiency gains do not trade off against the model's ability to handle object counting, spatial relations, or complex prompting.
>
> ---
>
> **Table 3. GenEval Results.**
>
> | Model | Overall (↑) | Single Obj. (↑) | Two Obj. (↑) | Counting (↑) | Colors (↑) | Position (↑) | GPU Hours (↓) |
> | :--- | :---: | :---: | :---: | :---: | :---: | :---: | :---: |
> | **SD3.5-M** | 0.63 | 0.98 | 0.78 | 0.50 | 0.81 | 0.24 | - |
> | **FlowGRPO** | 0.86 | 0.99 | 0.95 | 0.89 | 0.88 | 0.83 | 2000 |
> | **FlowGRPO (reduced)** | 0.73 | 0.99 | 0.85 | 0.77 | 0.81 | 0.67 | 1000 |
> | **BranchGRPO** | **0.89** | **0.99** | **0.96** | **0.90** | **0.89** | **0.85** | **1460** |

---

> > ### Author Response · Authors · 2025-11-20
> > **Reply to questions**
> >
> > **Q1. Are the reported KID/FID values sufficiently small to justify that BranchGRPO matches DanceGRPO in diversity?**
> >
> > This is an excellent point. We agree that distribution-level metrics like KID can mask conditional collapse.
> > To provide a more rigorous answer, we computed **prompt-conditioned diversity metrics: LPIPS-MPD and TCE**, following Kim et al. [1].
> >
> > As shown in Table 4, the diversity scores of BranchGRPO are remarkably close to the baseline.
> > Specifically, **LPIPS-MPD decreases by less than 0.6%** (0.723 → 0.719) compared to DanceGRPO.
> > Given the significant efficiency gains, this negligible variance confirms that structured branching preserves the diversity required for stable RLHF.
> >
> > ---
> >
> > **Table 4. Diversity metrics (higher is better).**
> >
> > | Method | LPIPS MPD | TCE |
> > | :--- | :---: | :---: |
> > | DanceGRPO | 0.723 | 4.45 |
> > | BranchGRPO (0,2,4,8) | 0.719 | 4.44 |
> > | BranchGRPO (0,3,6,9) | 0.713 | 4.42 |
> > | BranchGRPO (0,4,8,12) | 0.704 | 4.39 |
> >
> > These values show that BranchGRPO maintains both conditional and global diversity at the same level as DanceGRPO.
> >
> > [1] Kim et al., *Test-time Alignment of Diffusion Models without Reward Over-optimization*, ICLR 2025.
> >
> > ---
> >
> > **Q2. Does branching stage affect generational diversity?**
> >
> > Thank you for this insightful question regarding the branching stage.
> > The reviewer’s intuition is correct, and our measurements validate it:
> >
> > * **Early Branching** (e.g., {0,2,4,8}):
> >   Slightly higher LPIPS-MPD (0.719), reflecting larger variation because early timesteps govern global structure.
> >
> > * **Late Branching** (e.g., {0,4,8,12}):
> >   A small decrease (0.704), consistent with operating mainly on low-level refinements.
> >
> > Crucially, all variants remain very close to the sequential baseline (0.723), confirming that our default branching schedule strikes an optimal balance between diversity and efficiency.

---

### Official Review · Reviewer_UFyj · 2025-11-03

**Soundness:** 3
**Presentation:** 3
**Contribution:** 3
**Rating:** 6
**Confidence:** 3

**Summary:**

This paper redesigns GRPO for diffusion models into a branching rollout (BranchGRPO) that amortizes computation via shared prefixes, converts sparse terminal rewards into dense stepwise signals through leaf-reward fusion with depth-wise advantages, and reduces only the backprop cost via width/depth pruning. In experiments on image and video alignment, it trains faster and more stably than prior work (e.g., DanceGRPO), improves metrics such as HPS v2.1, and substantially shortens iteration time while maintaining or improving diversity and temporal consistency.

**Strengths:**

- The paper is well organized and easy to follow: the problem setup, methodology, and objectives are presented coherently, with clear sectioning and notation that make the technical ideas accessible.
- By redesigning sequential rollouts into branching rollouts, it amortizes computation via shared prefixes while preserving search breadth, which has novelty.
- With selective backpropagation (width/depth pruning) that keeps forward generation and evaluation intact while reducing only the learning compute, it enables flexible speed–quality trade-offs, so it has usefulness.

**Weaknesses:**

The proposed method introduces multiple new degrees of design freedom, e.g., branching positions/counts, branching factor, branch-noise correlation, reward-fusion temperature, pruning choice, depth window and its sliding schedule, and the Hybrid ODE–SDE mixing ratio. Although the paper includes ablations of these components, it does not provide practical guidance for selecting them when tasks or compute budgets change. As a result, there is a concern about benchmark-specific overfitting and variability in reproducibility.

**Questions:**

Please address the above concerns with concrete clarifications.

---

> ### Author Response · Authors · 2025-11-20
>
> We sincerely thank Reviewer UFyj for the constructive feedback. We appreciate your important concern regarding design freedom and reproducibility, which we address below with clear practical guidance.
>
> ---
>
> **W1. Concerns about design freedom, overfitting, and reproducibility.**
>
> We appreciate the reviewer’s valid concern regarding design freedom. We understand that introducing a branching structure adds hyperparameters. However, our extensive ablations (Sec. 4.4 / Fig. 5) confirm that these correspond to **stable structural trade-offs** (e.g., exploration width vs. compute) rather than sensitive tuning parameters.
>
> To directly address the **overfitting concern**, we emphasize that we used the **exact same default configuration** (Table 1) for **ALL** main experiments in the paper, spanning diverse architectures (FLUX.1-Dev, SD3.5-M) and modalities (WanX Video).
>
> To further guide practitioners, we identify two optimal configurations depending on resource priorities:
>
> ---
>
> ### **Scenario 1: Maximum Performance (Scaling Mode)**
>
> If the goal is to push the upper bound of alignment performance, we recommend using **BranchGRPO-Mix** and scaling the group size to **N = 81**
> (e.g., Branch Factor K = 3, Number of Branching Steps = 4).
>
> * **Benefit:**
>   Due to the efficiency of BranchGRPO-Mix, this configuration maintains iteration time and GPU-hour cost similar to DanceGRPO/FlowGRPO at N=16.
>
> * **Outcome:**
>   Under a similar compute budget, it yields **significant performance gains**
>   (HPS 0.360 → 0.387) by leveraging massive exploration that is computationally infeasible for sequential baselines.
>
> ---
>
> ### **Scenario 2: Balanced Efficiency & Quality (Universal Default)**
>
> If the goal is to accelerate training while maintaining or slightly improving quality, we recommend **BranchGRPO-DepthPruning** with the default settings in Table 1.
>
> * **Benefit:**
>   Delivers a **2.2× speedup** (314s vs. 698s per iteration) compared to the baseline.
>
> * **Outcome:**
>   Achieves strictly better alignment
>   (HPS 0.360 → 0.369)
>   without harming diversity or model capacity.
>
> We have consolidated these recommendations into Appendix A of the revised manuscript to ensure reproducibility.
>
> ---
>
> **Table 1. Recommended Universal Hyperparameter Settings.**
> We applied this single configuration across all tasks (T2I, I2V) and backbones (Flux, SD3.5, WanX) without task-specific tuning.
>
> | Component | Parameter | Default Value | Stable Range |
> | :--- | :--- | :---: | :---: |
> | **Branching** | Branch points | (0,3,6,9) | – |
> |  | Branching factor | 2 | {2, 3, 4} |
> | **Noise Correlation** | Correlation scale (s) | 4.0 | [3.0–5.0] |
> | **Reward Fusion** | Temperature (β) | 0 | {0, 1} |
> | **Pruning** | Pruning type | Depth Pruning | Depth or Width |
> |  | Depth window size | 5 | – |
> | **Hybrid ODE–SDE** | Mix ratio | 30% | [20%, 40%] |
>
> ---
>
> We commit to releasing the training code and configuration files to ensure full reproducibility.

---

### Author Response · Authors · 2025-11-20
**General Response**

We sincerely thank all reviewers for their constructive and insightful feedback.
Across the reviews, there is clear and consistent agreement on our core contributions:

(1) **Clarity of motivation and presentation** — highlighted by `Reviewer gDg6` and `Reviewer UFyj`,
(2) **Technical soundness of the methodology** — emphasized by `Reviewer gDg6` and `Reviewer gfya`,
(3) **Substantial improvements in efficiency, stability, and alignment quality** — noted by `Reviewer gDg6`, `Reviewer gfya`, and `Reviewer UFyj`,
(4) **Overall writing quality and readability** — praised by `Reviewer gDg6` and `Reviewer UFyj`.

During the rebuttal period, we conducted extensive additional experiments and analyses to address all remaining concerns. Below we summarize the major improvements incorporated into the revised submission.

---

# Summary of Revisions

We sincerely thank all reviewers again for their feedback, which has significantly strengthened our work.
The revised paper incorporates several key additions:

---

## 1. Expanded Generality and Applications

* Added validation on modern backbones: **SD3.5-M** and **Qwen-Image** (Section 4.3, Appendix C.1)
* Integrated additional GRPO-style baselines: **FlowGRPO** and **DiffusionNFT** (Section 4.3, Appendix C.1)
* Extended evaluation using the **GenEval benchmark** for compositionality and spatial understanding (Section 4.3, Appendix C.1)

---

## 2. Comprehensive Evaluation and Analysis

* Conducted a **human A/B preference study**, confirming that HPS-v2.1 gains reflect real human preference (Appendix C.1)
* Added **prompt-conditioned diversity metrics** (LPIPS-MPD, TCE) following Kim et al. (ICLR 2025) (Section 4.3)
* Provided **Reward–KL Pareto plots** and **CLIPScore** analysis to explain adherence trade-offs
  (Section 4.3, Appendix C.1)
* Added a **Best-of-N (BoN) reward scaling experiment**, demonstrating preserved generative diversity after finetuning (Appendix C.1)

---

## 3. Efficiency and Scalability Validation

* Added **strict Pareto efficiency comparisons** showing improved HPS while reducing compute (Section 4.3)
* Demonstrated **scalability to large group sizes (N = 81)** where DanceGRPO becomes infeasible (Section 4.5)

---

## 4. Enhanced Reproducibility and Practicality

* Added clear **hyperparameter selection guidelines** and recommendations (Appendix A)
* Provided a **unified default configuration** used across all experiments (Appendix A)

---

We believe these revisions fully address the reviewers’ concerns and significantly improve the clarity, robustness, and practical value of the paper.
All updates in the revised manuscript are highlighted for easy reference.

---

### Meta-Review · Area_Chair_gKkj · 2026-01-08

**Summary:**

This paper proposes BranchGRPO, a tree-structured branching rollout framework for GRPO-style RLHF in diffusion/flow-matching generative models. The method amortizes rollout cost via shared prefixes, reduces backward compute via pruning, and mitigates sparse terminal-reward credit assignment through reward fusion + depth-wise advantages that provide dense step-wise learning signals.
Key reviewer concerns included (i) whether performance gains are meaningful beyond runtime, and (ii) potential diversity degradation due to structured sampling/branching. The rebuttal adds substantial new experiments (human evaluation, broader metrics, additional backbones/baselines, diversity and adherence analyses), addressing most concerns empirically. Based on the reviews and post-rebuttal discussions, the AC recommends acceptance of the paper.

**Reviewer Concerns:**

Addressed by the rebuttal
- Hyperparameter guidance & reproducibility: The authors provide a unified default configuration and scenario-based recommendations (performance scaling vs. balanced efficiency), and demonstrate the same configuration across multiple backbones/modalities, reducing concerns about benchmark-specific tuning.
- Performance/impact concerns: While raw metric deltas can look modest, the rebuttal shows compute-matched or lower-compute comparisons where BranchGRPO achieves higher alignment, supported by human A/B preference results and multiple alignment metrics (HPS, PickScore, ImageReward, GenEval), alleviating concerns about over-optimizing a single proxy reward. Given consistent improvements under less resource, it is hard to characterize the gains as negligible.
- Diversity concerns: Beyond distributional proximity (KID/FID), the authors add prompt-conditioned diversity metrics (LPIPS-MPD, TCE) and Best-of-N scaling, with results close to baselines and no evidence of mode collapse.

Partially outstanding
- One reviewer felt that the absolute improvements in the final metrics were not large. However, the proposed method consistently achieves higher performance than the baselines while using substantially fewer computational resources. Achieving better alignment under reduced compute should not be considered a minor improvement; rather, it reflects a genuine performance gain enabled by the method itself, not merely a runtime optimization.
- Prompt adherence drift is observed, but analyses suggest it stems primarily from the reward model choice (HPS) rather than the branching mechanism; Reward–KL and CLIPScore analyses indicate BranchGRPO does not introduce additional drift beyond baseline behavior under the same reward.

**Reviewer Scores:**

If the reviewers had been able to participate fully in the discussion, the expected score changes would be better characterized as 6, 8, 4, 4 → (6 or 8), (8 or 10), 6, 4.

For Reviewer UFyj, the identified weakness was relatively limited and focused mainly on concerns about design freedom and reproducibility. These points were directly addressed with clear and concrete responses, including a unified default configuration used across all experiments, scenario-based recommended settings, and explicit hyperparameter ranges. Given that the concerns were not extensive and were well resolved, it is likely that the reviewer would maintain the original score or raise it (i.e., 6 → 6 or 8).

For Reviewer gDg6, the review was already strongly positive. The reviewer suggested additional experiments to strengthen generality and evaluation breadth, all of which were conducted during rebuttal, with results showing consistent trends across backbones, benchmarks, and settings. Since the original score was already high, it would most likely be maintained, with some possibility of a further increase after considering the expanded evidence (i.e., 8 → 8 or 10).
For Reviewer gfya, the reviewer explicitly stated a willingness to raise the score if the remaining concerns were resolved. The authors provided targeted additional results addressing those concerns, including prompt-conditioned diversity metrics, a human A/B preference study, broader backbone and baseline coverage, and Best-of-N scaling analyses. As these directly respond to the reviewer’s questions, an upward revision of the score appears likely (4 → 6).
Finally, Reviewer H6yd consistently maintained the original position throughout the discussion and explicitly chose to retain the initial rating after the rebuttal. Therefore, the score is expected to remain unchanged (4 → 4).

---

### Decision · Program_Chairs · 2026-01-26

Accept (Poster)